# Construction of new polygon mesh-type phantoms based on adult Japanese voxel phantoms

**Kaoru Sato** *, **Takuya Furuta**, **Daiki Satoh**, **Shuichi Tsuda**

Nuclear Science and Engineering Center, Japan Atomic Energy Agency, Tokai-mura, Ibaraki-ken, Japan

* sato.kaoru@jaea.go.jp

## Abstract

To construct adult Japanese phantoms applicable to individual exposure dose assessments, we created adult Japanese polygon mesh-type male (JPM) and female (JPF) phantoms through modification of the adult Japanese voxel phantoms, JM-103 (male) and JF-103 (female). The body sizes and masses of organs, tissues, and organ contents in JPM and JPF were adjusted to the Japanese averages, except for those unimportant for radiation protection or risks. The JPM and JPF data were converted to tetrahedral mesh-type data and incorporated into the Particle and Heavy Ion Transport code System (PHITS) for dose calculations. The dosimetric characteristics of the JPM and JPF phantoms were validated by calculating their effective doses in the anterior–posterior geometry for the external irradiation of photons with energies of 0.01–20 MeV and compared with those of JM-103 and JF-103 or the reference values given in ICRP Publication 116. The results confirmed no problems applying JPM and JPF to dose assessments in adult Japanese subjects. Furthermore, it was found that JPM and JPF can also accurately calculate the absorbed doses for entire organs and high radiosensitive cell regions with thin, small, and complicated structures.

## 1 Introduction

Quantitatively evaluating the absorbed doses to organs is necessary for estimating the radiation risks from radiation exposures [1]. Since organ doses cannot be directly measured or accurately evaluated in a living person, they have been evaluated using a combination of computational human phantoms and radiation transport simulation codes. The International Commission on Radiological Protection (ICRP) developed the reference computational phantoms of adult male (RCP-AM) and female (RCP-AF) [2] based on physiological and anatomical data of standard adult Caucasians [3]. The RCP-AM and RCP-AF are voxel phantoms comprising an aggregate of small rectangular block units called a volume pixel (voxel). The ICRP used the two reference voxel phantoms to calculate the dose conversion coefficients for external radiation fields [4] and the specific absorbed fractions for the intake of radionuclides [5].

Discussions are underway to reflect the purposes of the current 2007 Recommendations [1] in Japan's radiation safety regulations instead of the preceding 1990 Recommendations

**Data Availability Statement:** Except for our constructed the JPM and JPF phantom data (including the input files for implementing these phantoms in PHITS code), all relevant data are within the manuscript and its Supporting

information files. After acceptance for publication of our manuscript, these phantom data will be freely available from the GitHub repository (https://github.com/JapanesePolygonPhantom). The author's affiliation (Japan Atomic Energy Agency) is currently preparing to make the phantom data publicly available on the above GitHub repository. The publication preparation at the author's institution are proceeding without problems. This preparing process by the author's affiliation is almost complete. Therefore, the phantom data on repository of GitHub will be available at the same time as the date our paper is published on the PLOS ONE webpage.

**Funding:** The author(s) received no specific funding for this work.

**Competing interests:** The authors have declared that no competing interests exist.

[6] by the ICRP. Adult Japanese body sizes and organ masses are typically smaller than adult Caucasians [7]. These anatomical differences affect exposure doses. Therefore, it was necessary to develop phantoms to evaluate exposure doses considering the Japanese population's body sizes and organ masses. Thus, the authors first developed Japanese male (JM-103) and female (JF-103) voxel phantoms with the body sizes and organ masses [7] of an average adult Japanese [8]. The JM-103 and JF-103 phantoms also apply to dose assessment considering the tissue weighting factors ($w_T$) of ICRP Publication 103. And then, we developed a method to systematically model various body sizes for adult Japanese males and females by modifying JM-103 and JF-103. The created adult Japanese voxel phantom family can be applied to dose assessments against subjects within body size ranges covering more than 95% of the adult Japanese population. In comparisons between adult Japanese and Caucasian, the differences in masses, shapes, and locations of organs and tissues have a stronger impact on organ doses than differences in body sizes [8–11]. These phantoms have been used in diverse dose analyses [9, 12–18].

In individual exposures for medical treatments and radiological accidents, the exposure conditions, such as the individual's physical features at the time of exposure (e.g., postures and body sizes), irradiated parts (e.g., partial or whole-body), and radiation fields (e.g., radiation incident direction and energy), differ for each case. Therefore, more specific information on individuals and exposure conditions is needed for accurate dose assessments. However, voxel phantoms, which reproduce the shapes of organs and tissues by combining voxels of the same size, are challenging to use directly to calculate exposure doses considering the postures during exposure because of the limitations of the representation and resolution of a voxel format. Furthermore, small and complicated tissues (e.g., the lens of eyes, hereafter lens) and very thin tissues (e.g., the high radiosensitive cell regions in the respiratory and alimentary tracts, hereafter sensitive region) cannot be represented in the voxel phantoms because of the limited resolutions of voxel and original medical tomographic image data [2, 19].

The non-uniform rational B-spline and polygon surface modeling techniques have recently been employed to construct highly flexible and deformable human phantoms [20–30]. Surface modeling techniques enable us to accurately and conveniently represent an individual's physical features and very small or thin tissues. It offers more accurate dose assessments in target regions on radiation protection and radiation effect, and considerations for diverse exposure conditions during radiation work. The ICRP adopted the technique and newly provided the mesh-type reference computational phantom-adult male (MRCP-AM) and mesh-type reference computational phantom-adult female (MRCP-AF) which were defined by polygon or tetrahedral formats [19]. The MRCP-AM and MRCP-AF phantoms can reproduce the microstructures of small, thin, and complicated sensitive regions [1, 19, 31], such as the basal cell layer of membranous tissues and the lens, and calculate the doses absorbed by these sensitive regions. However, the description format of existing adult Japanese phantoms is only a voxel type.

Thus, this study aimed to construct Japanese polygon mesh-type male (JPM) and female (JPF) phantoms of adult Japanese based on JM-103 and JF-103. To validate the dosimetric characteristics of JPM and JPF, the effective doses of these polygon mesh-type Japanese phantoms were evaluated, and were compared with those of voxel phantoms, JM-103 and JF-103. The absorbed doses to the skin and eye tissues of the JPM and JPF phantoms, which contain structures that cannot be expressed in voxel phantoms, were then calculated when irradiated with photons or electrons to assess the reproduction of such small and thin regions. JPM and JPF are available on the GitHub repository website (https://github.com/JapanesePolygonPhantom).

## 2 Materials and methods

### 2.1 Adult Japanese voxel phantoms, JM-103 and JF-103

Tanaka and Kawamura [7] tabulated the systematic information on anatomical characteristics containing body size and organ mass of the Japanese on the basis of the measured values in autopsy cases during the period between 1970 and 1980. Autopsy was carried out at the Tokyo Metropolitan Government Medical Examiner's Office from 12 to 24 hours after death for normal subjects who died of sudden death. Thus, the Japanese averages of body size and organ mass reported by Tanaka and Kawamura [7] were obtained from the individuals regarded as practically normal and healthy, and were most appropriate for use in considering average Japanese phantom construction for the radiation protection purposes. Adult Japanese are typically shorter in height and lighter in body weight than adult Caucasians [3, 7]. Reflecting the difference in body size, the masses of most organs, tissues, and organ contents in adult Japanese are also smaller than in adult Caucasians. The JM-103 and JF-103 phantoms are voxel phantoms reproducing the above characteristics of adult Japanese [8]. The JM-103 and JF-103 are shown in Fig 1. JM-103 and JF-103 were constructed by modifying the JM and JF phantoms [11, 32, 33] based on computed tomography (CT) images of adult Japanese volunteers. The heights and weights of JM-103 and JF-103 were almost the same as the Japanese averages [8]. The masses of organs and organ contents of JM-103 and JF-103 were adjusted to the Japanese averages within ±10%, except for the bone tissue. Therefore, organ doses for representative adult Japanese can be evaluated using JM-103 and JF-103 [8]. Furthermore, their organs and tissues are segmented according to the tissue weighting factors ($w_T$) of the 2007 Recommendations [1]. The voxel size ($0.98 \times 0.98 \times 1.0$ mm$^3$) of JM-103 and JF-103 enables us to model the shapes of most organs and tissues, except for micrometer-sized structures such as sensitive regions and trabecular bones.

### 2.2 Construction of polygon mesh-type phantoms, JPM and JPF

**2.2.1 Extractions of surface shape data of organs and the body.**   The polygon mesh adopted for the JPM and JPF phantoms is a three-dimensional computer graphics image dataset comprising only surface shape data defined by vertices, edges, and faces. The American Standard Code for Information Interchange (ASCII) format segmentation image data [8] in the JM-103 and JF-103 phantoms were to construct the surface shape data of organs and tissues for JPM and JPF. First, the segmentation image data in JM-103 and JF-103 were converted from ASCII format to Tagged Image File Format (TIFF) using the mathematical and theoretical image-processing software Visilog 6.836 (FEI Company, Hillsboro, OR, USA). Then, the surface shape data of organs, tissues, and whole bodies were extracted from the TIFF segmentation image data of JM-103 and JF-103 using the rendering functions of vector-based three-dimensional imaging, modeling, and measurement software, 3D-DOCTOR 5.0 (Able Software Corp, Lexington, MA, USA), and stored as Wavefront Object (OBJ) format polygon mesh data.

**2.2.2 Modifications to the polygon mesh-type data of JPM and JPF.**   Sometimes, the polygon mesh-type data of organs, tissues, and bodies extracted from TIFF segmentation image data of JM-103 and JF-103 might fuse or overlap. Furthermore, intersections (polygons overlapping), holes, roughness on surfaces, and divisions into multiple parts were found in each polygon mesh-type data of the extracted organs and body. The above irregular structures are obstacles to applying polygon mesh-type data to constructing microscopic sensitive regions, tetrahedral mesh-type data, and radiation transport simulations. Therefore, the molding, scaling, and rearranging of polygon mesh-type data of organs and bodies were performed using the image-processing functions (scaling, rotation, moving, separation, smoothing,

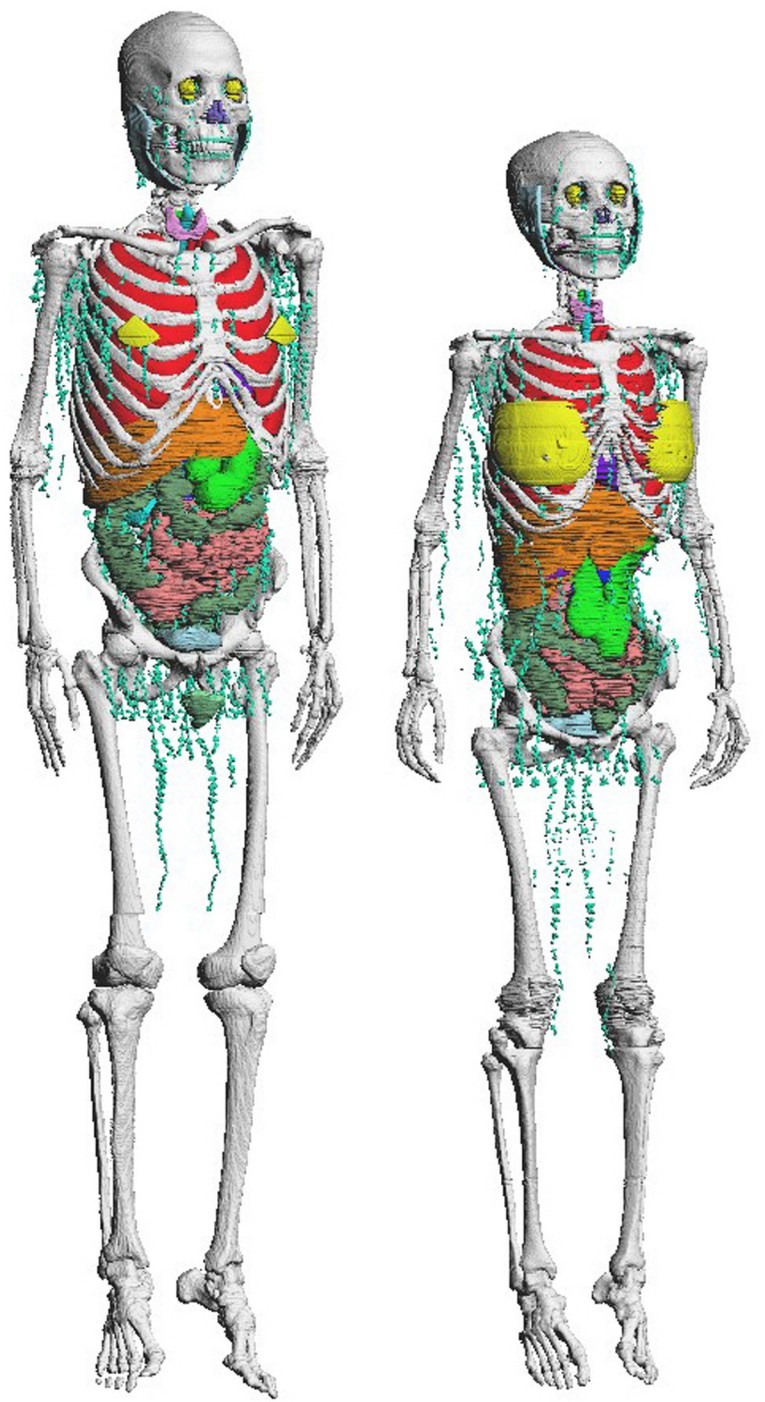

**Fig 1. The voxel-type Japanese phantoms of average adult male (left: JM-103) and average adult female (right: JF-103).** The skin, muscle, and adipose tissue are not displayed in this figure.

merging, and hole fills) of the three-dimensional computer graphics modeling software Meta-sequoia 4.8.6.a (Tetraface Inc, Shibuya-ku, Tokyo, Japan) and 3D-DOCTOR 5.0.

**2.2.3 Definition of sensitive regions.** The sensitive regions for membranous tissues, such as the respiratory tracts, alimentary tracts, urinary bladder, and skin, were modeled using the

functions of thickening and separation in Metasequoia 4.8.6.a. The thicknesses and structures of the membranous sensitive regions were referred to in ICRP Publications 66 [34], 100 [31], 103 [1], and 145 [19]. In the respiratory tracts of JPM and JPF, the extrathoracic regions (ET) containing $ET_1$ (anterior nose) and $ET_2$ (the posterior nasal passages, larynx, pharynx, and mouth), and the bronchial region, BB (trachea and bronchi), were hierarchized by image processing. One of the layers was defined as a sensitive region containing basal cells and secretory cells. Similar to the respiratory tracts, the sensitive regions containing epithelial or basal stem cells were also defined for the alimentary tracts (oral mucosa, esophagus stomach, small intestine, and colon) and urinary bladder. Where average mass data exist for adult Japanese, the masses of the membrane and contents were also adjusted to the Japanese averages [7] since the masses of organs and contents affect their geometries.

To match the average skin masses [7] of adult Japanese, the entire skin tissue was defined as the area from the outermost layer to depths of 1.28 mm (male) and 1.13 mm (female) in the whole body of JPM and JPF, respectively. The skin tissue thicknesses in JPM and JPF are close to the reference values in ICRP Publications 23 [35] and 89 [3]. The sensitive regions containing the basal cell layer in the skin were placed at depths ranging from 50 to 100 μm below the skin surface using the thickening function of Metasequoia 4.8.6.a.

The ICRP adopted the detailed eye tissue model of Behrens et al. [36] to calculate the dose conversion coefficients of the lenses [4]. In detailed eye tissue models, the eyeballs and lenses were realistically modeled. In particular, the lens comprises a radiosensitive region (hereafter the sensitive region) and another (hereafter the insensitive region). The anterior and posterior parts of the lens correspond to the sensitive and insensitive regions. Nguyen et al. [37] incorporated the detailed eye tissue model into MRCP-AM and MRCP-AF. This study also adopted the above eye tissue model to calculate the absorbed doses to the lenses for adult Japanese subjects. However, the average masses of eyeballs and lenses in adult Japanese males and females reported by Tanaka and Kawamura [7] were 15 g and 12 g, and 0.4 g and 0.3 g, respectively, and were lighter than those [2, 19] in adult Caucasians. Thus, the above detailed eye tissue model was adjusted to the average masses of adult Japanese by scaling, and was incorporated into JPM and JPF.

**2.2.4 Construction of the whole-body model of polygon mesh-type JPM and JPF.** To construct the whole-body model, polygon mesh data of each organ and tissue of JPM and JPF were integrated using image-processing functions. Then, for JPM and JPF, the existence of polygon intersections and holes, which are obstacles in radiation transport simulation based on the Monte Carlo method, was evaluated using the detection function of Metasequoia 4.8.6. a. If detected, they were improved with image processing. Figs 2 and 3 show the JPM and JPF phantoms, respectively.

**2.2.5 Conversions of the polygon mesh type to the tetrahedral mesh type in JPM and JPF.** Metasequoia 4.8.6.a edited the tag information relevant to object groups and material library templates of the constructed OBJ format polygon mesh-type data file to add information on the organ identification number used for JPM and JPF. The OBJ format polygon mesh-type data of JPM and JPF were converted to tetrahedral mesh-type data using POLY2-TET developed by Han et al. [38]. The generated tetrahedral mesh-type data comprises an ELE format file describing the relationship between the points constituting the tetrahedral elements and the organ identification numbers and a NODE format file describing the coordinate data of the points constituting the tetrahedral elements. The ELE and NODE format data files can be read using radiation transport calculation codes and used for dose analysis [38].

**2.2.6 Elemental composition and density of JPM and JPF.** The elemental compositions and densities of most organs and tissues in JPM and JPF were obtained from the reference data in ICRP Publication 89 [3] and ICRU Reports 44 [39] and 46 [40]. The elemental

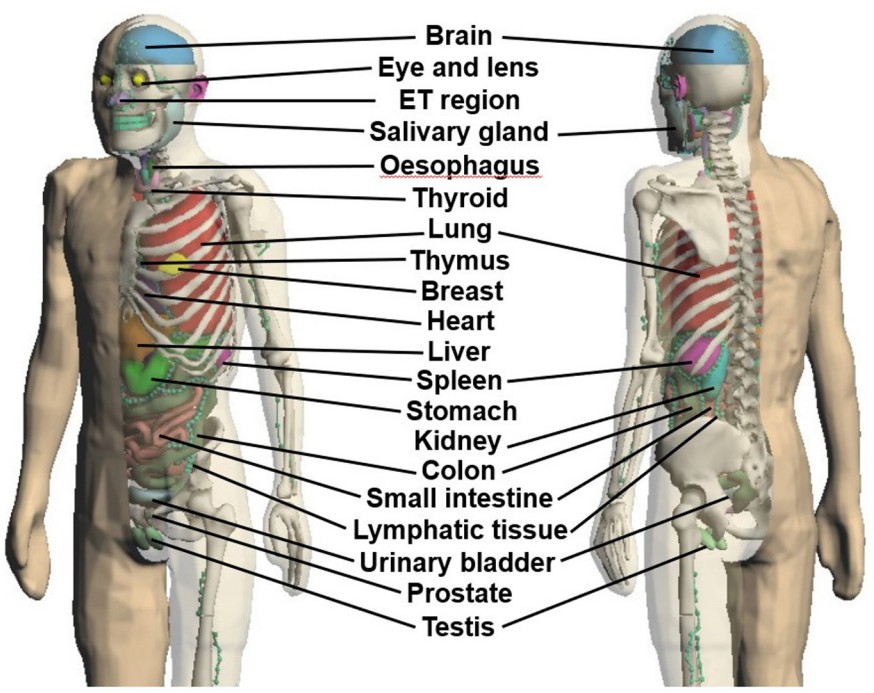

**Fig 2. Adult Japanese polygon mesh-type male (JPM) phantom.**

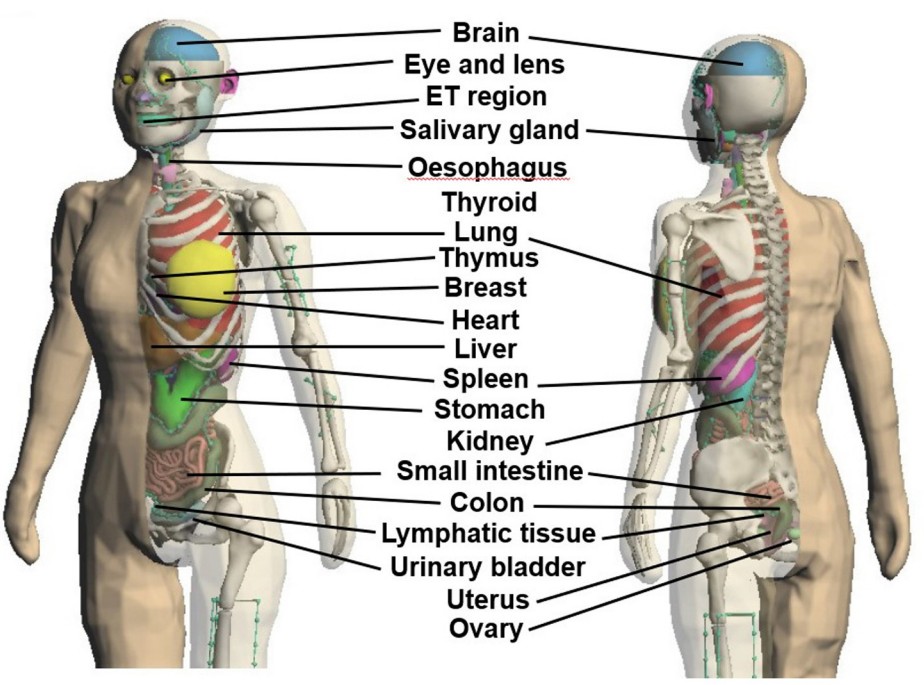

**Fig 3. Adult Japanese polygon mesh-type female (JPF) phantom.**

compositions and densities of the lymphatic tissue and prostate were referred to ICRP Publication 110 [2]. The elemental compositions and densities reported by ICRP Publication 145 [19] were used for the lenses and eyeballs. The tooth density reported by Schlattl et al. [41] was adopted for JPM and JPF. The width of the cavity of the trabecular bone in which the active marrow exists was very small, ranging from 0.2 to 1.7 mm [42]. Thus, it is challenging to accurately distinguish trabecular bone from active marrow on CT images. In addition, we are currently developing the functions for changing postures and body sizes of JPM and JPF, which is important for evaluating individual exposure doses. In changes of posture or body size, it is necessary to simultaneously change the shape of the anatomical bone regions (cervical vertebrae, thoracic vertebrae, lumbar vertebrae and ribs, etc. which change shape depending on posture.) and their internal structures (cortical bone, spongiosa and marrow in cavity, etc.) for each deformation pattern. At present, when anatomical bone regions and their internal structures are simultaneously changed by using the developing change functions for posture and body size, the errors including polygon intersections and holes occurred. The errors are obstacles to tetrahedral element generations and radiation transport calculations. Thus, the deformed phantom cannot be directly applied to tetrahedral element generations and radiation transport calculations. To prevent the error from occurring, it is necessary to determine the basic bone internal structure of cortical bone, spongiosa and marrow cavity that can accommodate changes in posture and body size of JPM and JPF. The basic bone internal structures in each anatomical bone regions are under construction. We plan to add the basic bone internal structure to the anatomical bone regions of JPM and JPF in the near future. Therefore, the bone tissues of JPM and JPF were segmented into 20 anatomical bone regions based on the divisions in ICRP Publications 70 [42] and 89 [3], and do not have internal structural regions including cortical bone, spongiosa, and marrow in cavity. In this study, anatomical bone regions in the JPM and JPF phantoms are defined as a mixture of hard bone, and active and inactive marrows. The elemental compositions and densities of hard bone, and active and inactive marrows were obtained from the reported data by Veit et al. [43] and ICRU Report 44 [39]. The elemental composition and density of each anatomical bone region in JPM and JPF was reevaluated by considering the mass ratios of the seven materials [8] assigned into each anatomical bone region of JM-103 and JF-103.

## 2.3 Physical characteristics and organ masses of the JPM and JPF

Table 1 presents the physical characteristics of adult Japanese polygonal or tetrahedral mesh-type and voxel phantoms. The heights and weights of JPM and JPF were 170 cm and 65 kg,

**Table 1. Physical characteristics of the JPM, JPF, JM-103 and JF-103 phantoms.**

| Property | | JPM | JPF | JM-103 | JF-103 |
|---|---|---|---|---|---|
| **Gender** | | **male** | **female** | **male** | **female** |
| Age | | 54 | 54 | 54 | 54 |
| Height [cm] | | 170 (170)* | 155 (155) | 171 | 155 |
| Body weight [kg] | | 65 (64) | 52 (52) | 65 | 52 |
| Number of total segmented region | | 198 | 198 | 214 | 215 |
| Total data size of phantom [MB] | Polygon mesh | 59 | 66 | - | - |
| | Tetrahedral mesh | 244 | 280- | - | - |
| | Voxel | - | - | 1800 | 1700 |

* The values in parenthesis are the average heights and body weights [7] of adult Japanese males and females.

and 155 cm and 52 kg, respectively. Similar to the JM-103 and JF-103 phantoms, the heights and weights of JPM and JPF phantoms were matched to the Japanese averages [7]. The total segmented regions in JPM and JPF were 198 and 198, respectively.

The data sizes of the polygon mesh-type JPM and JPF are 59 and 66 MB, respectively, which is small compared to 244 and 280 MB in the tetrahedral mesh-type phantom data. Furthermore, they are significantly smaller than the data sizes of the two Japanese voxel phantoms. The difference in data sizes results from the difference in each phantom description format. The polygon mesh-type phantom data have the smallest size because it comprises only the object's surface shape data. However, the tetrahedral mesh-type and voxel phantoms have surface shape and internal structure data. Tetrahedral elements of nonuniform size and shape can accurately represent the internal structure of the tetrahedral phantom data. This representation method can minimize and optimize the data sizes. However, the voxel size must be sufficiently small to describe small internal structures, and voxels of uniform size and shape must represent the internal structures of voxel phantoms. Consequently, the voxel phantom data have the largest size among the three description types. The calculation time for organ doses depends on the numbers and data sizes of the voxels and elements used to represent the phantoms [44]. It can be expected that JPM and JPF will reduce the time required for dose calculations compared with JM-103 and JF-103.

As explained in section 2.1, the Japanese averages of organ mass were derived from the measured values in the autopsy. At autopsy, the masses of the organ and the blood in the organ were measured separately. The Japanese average of organ masses given by Tanaka and Kawamura [7] is the sum of organ mass and blood mass in organ. Thus, the organ masses of the voxel phantoms (JM-103 and JF-103) and polygon mesh-type phantoms (JPM and JPM) were defined to include blood in the organs. As shown in Tables 2 and 3, the masses of most organs, tissues, and organ contents in the JM-103 and JF-103 phantoms were modeled to be within approximately 10% of the average values for adult Japanese because of the limited voxel representation. However, the masses of organs, tissues, and organ contents of the JPM and JPF phantoms were perfectly adjusted to match the average values [7] of adult Japanese, except for adipose, hard bone, and inactive marrow, which are not crucial for dose assessments and have low radiosensitivity.

## 2.4 Distributions and dose calculations for bone marrow in JPM and JPF

In adults, bone tissue, such as the active and inactive marrows and endosteum, are not uniformly distributed in the body [2, 3, 19, 42]. Therefore, the distributions of the active and inactive marrows and endosteum in the body are necessary to calculate appropriate absorbed doses. Sato et al. [8] defined the fine distributions of the active and inactive marrows in whole-body skeleton tissues for the JM-103 and JF-103 phantoms. In JM-103 and JF-103, the 20 anatomical bone regions based on the divisions in ICRP Publications 70 [42] and 89 [3] were segmented from the whole-body skeleton tissues. Furthermore, according to the grey values, each anatomical bone region was resegmented into seven materials with different total bone marrow (active and inactive marrows) contents (%). The grey values indicate the attenuation coefficients, i.e., the electron density of pixels corresponding to each tissue on the CT image, and are closely related to the materials and tissue densities of organs and tissues. By these processes, the bone tissues of JM-103 and JF-103 were segmented into 140 regions (7 materials × 20 anatomical bone regions) over the whole body. In each anatomical region of JM-103 and JF-103, the complex internal structure of bone tissue including cortical bone, spongiosa and marrow cavity are approximately represented as the density distribution in bone region which were segmented based on the above image processing.

**Table 2. Masses of organs, tissues, and contents of JPM and JM-103 [8] and the averages [7] of adult Japanese males.**

| Organ, tissue and content | Mass [kg] | | | | |
|---|---|---|---|---|---|
| | JPM | | JM-103 | | Average of adult Japanese male |
| Adipose | 17.386 | (1.25)* | 14.192 | (1.02) | 13.900 |
| Adrenal | 0.014 | (1.00) | 0.015 | (1.04) | 0.014 |
| Brain | 1.470 | (1.00) | 1.529 | (1.04) | 1.470 |
| Breast (adipose + mammary glands) | 0.022 | (1.00) | 0.023 | (1.03) | 0.022 |
| Bronchi | 0.011 | (-) | 0.009 | (-) | - |
| Colon | 0.330 | (1.00) | 0.326 | (0.99) | 0.330 |
| Colon content | 0.360 | (1.00) | 0.360 | (1.00) | 0.360 |
| Endosteum | 0.466 | (-) | 0.574 | (-) | - |
| Esophagus | 0.040 | (1.00) | 0.036 | (0.91) | 0.040 |
| ET ($ET_1+ET_2$) regions | 0.004 | (-) | 0.043 | (-) | - |
| Eyeball | 0.015 | (1.00) | 0.014 | (0.92) | 0.015 |
| Gall bladder | 0.008 | (1.00) | 0.008 | (1.02) | 0.008 |
| Gall bladder content | 0.050 | (1.00) | 0.049 | (0.98) | 0.050 |
| Hard bone | 5.351 | (1.19) | 6.730 | (1.50) | 4.500 |
| Heart | 0.380 | (1.00) | 0.389 | (1.02) | 0.380 |
| Heart content | 0.400 | (1.00) | 0.362 | (0.91) | 0.400 |
| Kidney | 0.320 | (1.00) | 0.333 | (1.04) | 0.320 |
| Lens | 0.0004 | (1.00) | 0.0004 | (0.95) | 0.0004 |
| Liver | 1.600 | (1.00) | 1.462 | (0.91) | 1.600 |
| Lung | 1.200 | (1.00) | 1.215 | (1.01) | 1.200 |
| Lymphatic tissue | 0.220 | (1.00) | 0.224 | (1.02) | 0.220 |
| Marrow (active) | 1.001 | (1.00) | 1.192 | (1.19) | 1.000 |
| Marrow (inactive) | 2.121 | (1.63) | 2.526 | (1.94) | 1.300 |
| Muscle | 27.429 | (1.00) | 28.198 | (1.03) | 27.500 |
| Oral mucosa | 0.00015 | (-) | 0.010 | (-) | - |
| Pancreas | 0.130 | (1.00) | 0.136 | (1.05) | 0.130 |
| Prostate | 0.012 | (1.00) | 0.011 | (0.94) | 0.012 |
| Salivary gland | 0.082 | (1.00) | 0.086 | (1.05) | 0.082 |
| Skin | 2.411 | (1.00) | 2.189 | (0.91) | 2.400 |
| Small intestine | 0.590 | (1.00) | 0.557 | (0.94) | 0.590 |
| Small intestine content | 0.350 | (1.00) | 0.351 | (1.00) | 0.350 |
| Spleen | 0.140 | (1.00) | 0.139 | (1.00) | 0.140 |
| Stomach | 0.140 | (1.00) | 0.141 | (1.01) | 0.140 |
| Stomach content | 0.240 | (1.00) | 0.240 | (1.00) | 0.240 |
| Tooth | 0.059 | (1.30) | 0.061 | (1.35) | 0.045 |
| Testis | 0.037 | (1.00) | 0.036 | (0.99) | 0.037 |
| Thymus | 0.030 | (1.00) | 0.031 | (1.03) | 0.030 |
| Thyroid | 0.019 | (1.00) | 0.020 | (1.06) | 0.019 |
| Tongue | 0.067 | (1.00) | 0.062 | (0.92) | 0.067 |
| Trachea | 0.009 | (1.00) | 0.009 | (0.99) | 0.009 |
| Urinary bladder | 0.040 | (1.00) | 0.039 | (0.97) | 0.040 |
| Urinary bladder content | 0.100 | (1.00) | 0.102 | (1.02) | 0.100 |

Notes: Masses of organs and tissues of JPM and JM-103, and the Japanese averages includes the masses of blood in organs and tissues.

* The values in parenthesis are the mass ratios of JPM and JM-103 to averages of adult Japanese males.

**Table 3. Masses of organs, tissues, and contents of JPF and JF-103 [8] and the averages [7] of adult Japanese females.**

| Organ, tissue and content | Mass [kg] | | | | |
|---|---|---|---|---|---|
| | JPF | | JF-103 | | Average of adult Japanese female |
| Adipose | 15.048 | (0.99)* | 13.815 | (0.91) | 15.200 |
| Adrenal | 0.013 | (1.00) | 0.012 | (0.92) | 0.013 |
| Brain | 1.320 | (1.00) | 1.335 | (1.01) | 1.320 |
| Breast (adipose + mammary glands) | 0.300 | (1.00) | 0.309 | (1.03) | 0.300 |
| Bronchi | 0.012 | (-) | 0.020 | (-) | - |
| Colon | 0.260 | (1.00) | 0.244 | (0.94) | 0.260 |
| Colon content | 0.280 | (1.00) | 0.289 | (1.03) | 0.280 |
| Endosteum | 0.352 | (-) | 0.406 | (-) | - |
| Esophagus | 0.030 | (1.00) | 0.031 | (1.05) | 0.030 |
| ET ($ET_1$+$ET_2$) regions | 0.002 | (-) | 0.030 | (-) | - |
| Eyeball | 0.012 | (1.00) | 0.012 | (0.99) | 0.012 |
| Gall bladder | 0.006 | (1.00) | 0.006 | (1.03) | 0.006 |
| Gall bladder content | 0.038 | (1.00) | 0.035 | (0.92) | 0.038 |
| Hard bone | 4.027 | (1.18) | 4.460 | (1.31) | 3.400 |
| Heart | 0.300 | (1.00) | 0.325 | (1.08) | 0.300 |
| Heart content | 0.320 | (1.00) | 0.316 | (0.99) | 0.320 |
| Kidney | 0.280 | (1.00) | 0.271 | (0.97) | 0.280 |
| Lens | 0.0003 | (1.00) | 0.0003 | (0.94) | 0.0003 |
| Liver | 1.400 | (1.00) | 1.311 | (0.94) | 1.400 |
| Lung | 0.910 | (1.00) | 0.978 | (1.07) | 0.910 |
| Lymphatic tissue | 0.170 | (1.00) | 0.173 | (1.02) | 0.170 |
| Marrow (active) | 0.775 | (0.99) | 0.956 | (1.23) | 0.780 |
| Marrow (inactive) | 1.557 | (1.57) | 1.911 | (1.93) | 0.990 |
| Muscle | 20.704 | (1.00) | 20.212 | (0.97) | 20.790 |
| Oral mucosa | 0.0001 | (-) | 0.008 | (-) | - |
| Ovary | 0.011 | (1.00) | 0.012 | (1.09) | 0.011 |
| Pancreas | 0.110 | (1.00) | 0.113 | (1.02) | 0.110 |
| Salivary gland | 0.062 | (1.00) | 0.063 | (1.02) | 0.062 |
| Skin | 1.807 | (1.00) | 1.898 | (1.05) | 1.800 |
| Small intestine | 0.450 | (1.00) | 0.467 | (1.04) | 0.450 |
| Small intestine content | 0.270 | (1.00) | 0.283 | (1.05) | 0.270 |
| Spleen | 0.120 | (1.00) | 0.110 | (0.92) | 0.120 |
| Stomach | 0.110 | (1.00) | 0.106 | (0.97) | 0.110 |
| Stomach content | 0.180 | (1.00) | 0.180 | (1.00) | 0.180 |
| Tooth | 0.045 | (1.33) | 0.046 | (1.35) | 0.034 |
| Thymus | 0.029 | (1.00) | 0.028 | (0.96) | 0.029 |
| Thyroid | 0.017 | (1.00) | 0.017 | (0.99) | 0.017 |
| Tongue | 0.051 | (1.00) | 0.051 | (1.01) | 0.051 |
| Trachea | 0.007 | (1.00) | 0.007 | (0.97) | 0.007 |
| Urinary bladder | 0.030 | (1.00) | 0.032 | (1.06) | 0.030 |
| Urinary bladder content | 0.085 | (1.00) | 0.090 | (1.06) | 0.085 |
| Uterus | 0.070 | (1.00) | 0.067 | (0.96) | 0.070 |

Notes: Masses of organs and tissues of JPF and JF-103, and the Japanese averages includes the masses of blood in organs and tissues.

* The values in parenthesis are the mass ratios of JPF and JF-103 to averages of adult Japanese females.

As described in subsection 2.2.6, we are developing a method to change the posture and body size of JPM and JPF. However, the developing posture and body shape deformation methods currently face the challenge of simultaneously and reasonably deforming the outer surface shape of bone tissue and its internal structures including cortical bone, spongiosa and marrow in cavity. For this reason, the bone tissue regions of JPM and JPF were only divided into 20 anatomical bone regions, similar to JM-103 and JF-103, without segmenting these internal structural regions. Thus, the densities and elemental compositions of 20 anatomical bone regions in JPM and JPF were reevaluated by considering the mass ratios of the seven materials assigned into each anatomical region of JM-103 and JF-103.

Tables 4 and 5 show the masses of active and inactive marrows, hard bone, and endosteum of 20 anatomical bone regions evaluated for the JPM and JPF phantoms. The masses of the total bone marrow and hard bone in each anatomical bone region were determined over the whole-skeleton system, according to the bone component data of JM-103 and JF-103. The active marrow masses were allocated to the 20 anatomical bone regions based on the mass ratio data [3, 42] of active marrow in the whole-skeleton tissues of an adult. No research data exist on the mass distribution of the endosteum in adult Japanese individuals' whole-body skeleton tissues. Therefore, the mass ratios of the endosteum in the body of JPM and JPF were referred to those [2] of RCP-AM and RCP-AF.

As explained in Subsection 2.2.6, the bone tissues in the JPM and JPF phantoms were assumed to be a mixture of bone components, such as hard bone, active and inactive marrows, and endosteum. Thus, it was necessary to calculate the absorbed doses for each bone component separately. Therefore, the established method [45] was adopted to calculate the energy deposited on each bone component in a mixture of bone tissue in JPM and JPF. This method

**Table 4. Masses of active and inactive marrows, hard bone, and endosteum of each anatomical bone region in the JPM phantom.**

| Anatomical bone region | Mass [kg] | | | | |
|---|---|---|---|---|---|
| | Active marrow | Inactive marrow | Hard bone | Endosteum | Total |
| Humeri, upper half | 0.024 | 0.086 | 0.158 | 0.008 | 0.277 |
| Humeri, lower half | 0.000 | 0.067 | 0.149 | 0.010 | 0.226 |
| Ulnae and radii | 0.000 | 0.072 | 0.152 | 0.014 | 0.238 |
| Wrist and hand bones | 0.000 | 0.060 | 0.107 | 0.008 | 0.176 |
| Clavicles | 0.008 | 0.010 | 0.040 | 0.002 | 0.060 |
| Cranium | 0.077 | 0.224 | 1.059 | 0.073 | 1.432 |
| Femora, upper half | 0.071 | 0.132 | 0.382 | 0.039 | 0.623 |
| Femora, lower half | 0.000 | 0.292 | 0.444 | 0.027 | 0.763 |
| Tibiae, fibiae, patellae | 0.000 | 0.437 | 0.652 | 0.097 | 1.185 |
| Ankle and foot bones | 0.000 | 0.279 | 0.410 | 0.028 | 0.717 |
| Mandible | 0.007 | 0.028 | 0.128 | 0.002 | 0.165 |
| Pelvis | 0.185 | 0.152 | 0.451 | 0.045 | 0.834 |
| Ribs | 0.157 | 0.122 | 0.338 | 0.026 | 0.643 |
| Scapulae | 0.029 | 0.087 | 0.221 | 0.009 | 0.345 |
| Cervical vertebrae | 0.038 | 0.009 | 0.101 | 0.010 | 0.157 |
| Thoracic vertebrae | 0.157 | 0.005 | 0.218 | 0.024 | 0.404 |
| Lumbar vertebrae | 0.120 | 0.042 | 0.192 | 0.021 | 0.375 |
| Sacrum | 0.097 | 0.008 | 0.103 | 0.018 | 0.226 |
| Sternum | 0.031 | 0.009 | 0.044 | 0.005 | 0.088 |
| Os hyoideum | 0.001 | 0.000 | 0.002 | 0.000 | 0.003 |
| Total | 1.001 | 2.121 | 5.351 | 0.466 | 8.938 |

**Table 5. Masses of active and inactive marrows, hard bone, and endosteum of each anatomical bone region in the JPF phantom.**

| Anatomical bone region | Mass [kg] | | | | |
|---|---|---|---|---|---|
| | Active marrow | Inactive marrow | Hard bone | Endosteum | Total |
| Humeri, upper half | 0.019 | 0.071 | 0.135 | 0.006 | 0.231 |
| Humeri, lower half | 0.000 | 0.037 | 0.096 | 0.007 | 0.140 |
| Ulnae and radii | 0.000 | 0.059 | 0.113 | 0.010 | 0.183 |
| Wrist and hand bones | 0.000 | 0.029 | 0.035 | 0.006 | 0.070 |
| Clavicles | 0.006 | 0.005 | 0.022 | 0.002 | 0.035 |
| Cranium | 0.060 | 0.221 | 0.987 | 0.056 | 1.322 |
| Femora, upper half | 0.055 | 0.118 | 0.335 | 0.030 | 0.537 |
| Femora, lower half | 0.000 | 0.187 | 0.299 | 0.021 | 0.507 |
| Tibiae, fibiae, patellae | 0.000 | 0.357 | 0.527 | 0.073 | 0.957 |
| Ankle and foot bones | 0.000 | 0.177 | 0.245 | 0.021 | 0.443 |
| Mandible | 0.006 | 0.016 | 0.119 | 0.001 | 0.143 |
| Pelvis | 0.144 | 0.071 | 0.295 | 0.034 | 0.544 |
| Ribs | 0.122 | 0.075 | 0.237 | 0.020 | 0.454 |
| Scapulae | 0.022 | 0.042 | 0.122 | 0.007 | 0.193 |
| Cervical vertebrae | 0.030 | 0.003 | 0.057 | 0.008 | 0.097 |
| Thoracic vertebrae | 0.119 | 0.000 | 0.136 | 0.018 | 0.273 |
| Lumbar vertebrae | 0.093 | 0.025 | 0.133 | 0.016 | 0.267 |
| Sacrum | 0.075 | 0.053 | 0.105 | 0.014 | 0.247 |
| Sternum | 0.024 | 0.011 | 0.027 | 0.004 | 0.066 |
| Os hyoideum | 0.000 | 0.001 | 0.002 | 0.000 | 0.003 |
| Total | 0.775 | 1.557 | 4.027 | 0.352 | 6.711 |

helps evaluate energy deposition to the active marrow and endosteum of other phantoms with unique and various elemental compositions and densities, for which the dose-response functions [4] developed by ICRP for RCP-AM and RCP-AF are not applicable. The deposited energy, $ED_{\text{bone},a}$ (MeV), for photon exposures to the active marrow or endosteum of the anatomical bone region $a$ was calculated using Eq (1),

$$ED_{\text{bone},a} = \sum_a M_a \int_0^{E_p} \Phi_a(E) \frac{\mu_{en}}{\rho}(E)\, E\, \mathrm{d}E, \qquad (1)$$

where $M_a$ (kg) is the mass of the active marrow or endosteum in the anatomical bone region $a$, $E_p$ (MeV) is the maximum energy of photons passing through the anatomical bone region $a$, and $\Phi_a(E)$ (m$^{-2}$) is the incident particle fluence in the anatomical bone region $a$ for the photon with an energy of $E$ (MeV). $\mu_{en}/\rho$ $(E)$ (m$^2$ kg$^{-1}$) is the energy-dependent mass energy-absorption coefficients for the active marrow or endosteum which is calculated using the elemental composition data of JPM and JPF and the element-specific mass energy-absorption coefficients obtained from reports by Hubbell and Seltzer [46]. However, $ED_{\text{bone},a}$ is overestimated by the use of $\mu_{en}/\rho$, when the photon energy is more than a few MeV [47]. Therefore, the $ED_{\text{bone},a}$ values for photons with 1 MeV or more were calculated by dividing the total energy deposited in each anatomical bone region in accordance with the mass ratio of the active marrow and endosteum. On the other hand, for photon irradiation below 0.2 MeV, secondary electron equilibrium in bone tissue is not established. It should also be noted that in some cases under conditions of localized external or internal exposure, secondary electron equilibrium may not be established in the target region. Photoelectrons resulting from the

photoelectric effect are emitted from the bone trabeculae. The emitted photoelectrons cause the increase in the absorbed doses to the neighboring active marrow and endosteum [4, 48]. This dose enhancement effect to the active marrow and endosteum was not taken into Eq (1). Eq (1) might lead to underestimate the energy deposited in the active marrow and endosteum for irradiation of photons with 0.2 MeV or less. Unlike JM-103 and JF-103, each anatomic bone region in JPM and JPF is defined as a composite tissue which consists of the hard bone and bone marrow, and have not internal structures made up of bone tissue regions with different densities. Therefore, it is necessary to confirm whether the internal structures of bone tissue impact the absorbed doses to the active marrow and endosteum or not. The absorbed doses to the active marrow or endosteum of JPM and JPF at irradiation of photons with energy ranges from 0.01 to 20 MeV in AP geometry were compared with those of JM-103, JF-103, RCP-AM and RCP-AF which have non-uniform density distribution in each anatomical bone region (S1 Data). The absorbed doses to the active bone marrow and endosteum of JPM, JPF, JM-103 and JF-103 were evaluated using Eq (1) at all calculated energy ranges, instead of the above-mentioned mass ratio correction. In most photon energies, the absorbed doses to active marrow and endosteum in JPM and JPF were consistent with those in JM-103 and JF-103. However, the differences in absorbed doses to active marrow and endosteum between JPM and JM-103 or JPF and JF-103, at energy ranges from 0.01 to 0.05 MeV and from 15 to 20 MeV, were more than 10%. Photons incident on the bone tissue interact with complicated internal structures including hard bone and bone marrow, and their absorbed doses are affected by shielding or buildup. Therefore, the differences in absorbed dose to bone marrow and endosteum between polygon mesh-type (JPM and JPF) and voxel type (JM-103 and JF-103) phantoms are presumably due to the internal structure of anatomical bone regions.

At 0.01 and 0.015 MeV photon irradiation, the absorbed doses to the endosteum and active marrow were higher for Japanese voxel-type phantoms (JM-103 and JF-103), Japanese polygon mesh-type phantoms (JPM and JPF), and ICRP reference phantoms (RCP-AM and RCP-AF), in that order (S1 Data). This is due to differences in the internal structure of bone tissue; the outermost surface of the bone tissue of RCP-AM and RCP-AF is covered by cortical bone with high density. Therefore, low-energy photons are attenuated by the shielding effect of the cortical bone. As a result, the absorbed dose to the active marrow and endosteum located in the spongiosa and marrow cavity is reduced. Based on the bone density data obtained from the CT images of the actual persons employed as volunteers for the development of the two Japanese male (JM) and female (JF) voxel phantoms [11], seven materials with different densities and total bone marrow contents (%) were assigned into each voxel belonging to the bone tissue regions of JM-103 and JF-103. This indicates that both high- and low-density regions are located together in the outermost surface of the bone tissue. Low-energy photons are incident under conditions where the shielding by hard bones be negligible, and directly increase the absorbed dose in the endosteum and active marrow. Secondary electrons resulting from interactions with photons incident on voxel assigned hard bone rich material, deposit additionally energy in adjacent voxel which is defined bone marrow rich material, because the voxel size (about 1mm$^3$) of JM-103 and JF-103 is very small. It was suggested the difference in structures of bone tissue regions caused the elevation of the absorbed doses to the active marrow and endosteum of JM-103 and JF-103 compared with those of other phantoms.

For photon irradiation in the energy range of 0.02–0.3 MeV, the absorbed doses to the active marrow and endosteum in the Japanese phantoms (JPM, JPF, JM-103, and JF-103) were lower than those in RCP-AM and RCP-AF (S1 Data). The maximum difference was approximately 60%. These results supported facts that the microscopic energy deposition processes of secondary electrons resulting from their interaction with photons reported by ICRP Publication 116 [4] and Johnson et al [48] are not considered.

Absorbed doses ($D_a$) (J kg$^{-1}$) to the activity marrow or endosteum in the anatomical bone region $a$ were calculated using Eq (2),

$$D_a = \frac{ED_{\text{bone},a}\, c}{M_a},$$ (2)

where $c$ (J MeV$^{-1}$) is the constant value ($1.602 \times 10^{13}$) to convert the unit from MeV to J. The doses absorbed by the active marrow and endosteum over the whole body were calculated by mass-weighted averaging based on the mass data of the active marrow and endosteum listed in Tables 4 and 5.

## 2.5 Organ dose and effective dose calculations

**2.5.1 Incorporation of JPM, JPF, JM-103, and JF-103 into the PHITS code.** A general-purpose radiation transport simulation code, PHITS version 3.33 [49] was used for calculations of absorbed doses to organs, tissues and sensitive regions due to external photon or electron exposures. PHITS can read the voxel and tetrahedral mesh-type phantoms and simulate the analysis of radiation behaviors in the human body.

The organ-segmented data of the voxel phantoms, JM-103 and JF-103, are recorded in ASCII format. Each numeral in ASCII format data corresponds to the organ identification number assigned to each pixel. PHITS can read the repeating structure of a lattice function in General Geometry (GG) format, a computational geometry format. We converted the organ-segmented data of JM-103 and JF-103 to the GG format. The total data sizes of the GG format for JM-103 and JF-103 data are approximately 600 MB and 590 MB, respectively.

The JPM and JPF phantoms were converted from polygon mesh type to tetrahedral mesh type by the methods described in Subsection 2.2.5 and were incorporated into the PHITS code through its function [44] of reading tetrahedral geometry.

**2.5.2 Dose calculation conditions in the radiation transport simulation.** According to the idealized exposure conditions [4] defined by the ICRP, the organ absorbed doses or effective doses were calculated by assuming whole-body irradiation to uniform parallel photon or electron beam under the anterior-to-posterior (AP) geometry. The organ absorbed doses and effective doses were calculated for 33 incident photon or electron energies ranging from 0.01 to 20 MeV. The organ absorbed doses and effective doses are normalized to incident particle fluence and are given in pGy cm$^2$ and pSv cm$^2$, respectively. According to the methods of ICRP Publication 103 [1], the equivalent doses to organs and tissues in the male and female phantoms were calculated separately. The average values of the equivalent doses for the male and female phantoms were used to evaluate the sex-averaged effective doses.

In most calculation cases, the number of produced primary photons or electrons in the PHITS calculation was set to obtain fraction standard deviations of less than 1% in the deposited energy to each organ, tissue, and sensitive region. For the calculation cases of the doses absorbed in the sensitive region of lenses due to irradiating electrons below 0.5 MeV, the number of primary electrons was set to achieve fraction standard deviations of less than 5% because of the short ranges of electrons. The Electron Gamma Shower version 5 (EGS5) algorithm [50] embedded in the PHITS code was used to transport photons and electrons. The cut-off energy of photons and electrons was 1 keV for all calculations.

## 3 Results and discussion

### 3.1 Validations of the effective doses evaluated using JPM and JPF

Whether the JPM and JPF phantoms can be practically used for dose assessments was confirmed by evaluating the effective doses according to the methods explained in Subsection 2.5.2. Since the sensitive region was not segmented in the skin tissue of the JM-103 and JF-103 [8], the effective doses were derived using the equivalent dose in the entire skin (hereafter referred to as the all-skin region). Effective doses for the Japanese polygon mesh-type phantoms were calculated using the sex-averaged equivalent doses based on JPM and JPF. Similarly, the sex-averaged equivalent doses of JM-103 and JF-103 were used to calculate the effective doses for the Japanese voxel phantoms. Fig 4 compares the effective doses of Japanese polygon mesh-type phantoms by photon irradiation under AP geometry with those of the Japanese voxel phantoms and the reference values [4] of the ICRP (S1 Table). The effective doses of the Japanese polygon mesh-type phantoms agreed well with those of the Japanese voxel phantoms within approximately 5% at energies of 0.04 MeV or above. In the energy ranges from 0.01 to 0.03 MeV, the differences in effective doses between the Japanese phantoms modeled by the two description formats increased significantly with decreasing irradiation energy and ranged from -58% to -8%. The characteristics of the phantom description formats caused the differences in the effective doses. The membranous organs, such as the skin and alimentary tracts of the voxel phantoms, cannot fully enclose their contents because of limitations of resolution and geometry in a voxel format. The breast, muscle, and adipose tissue, located just below the

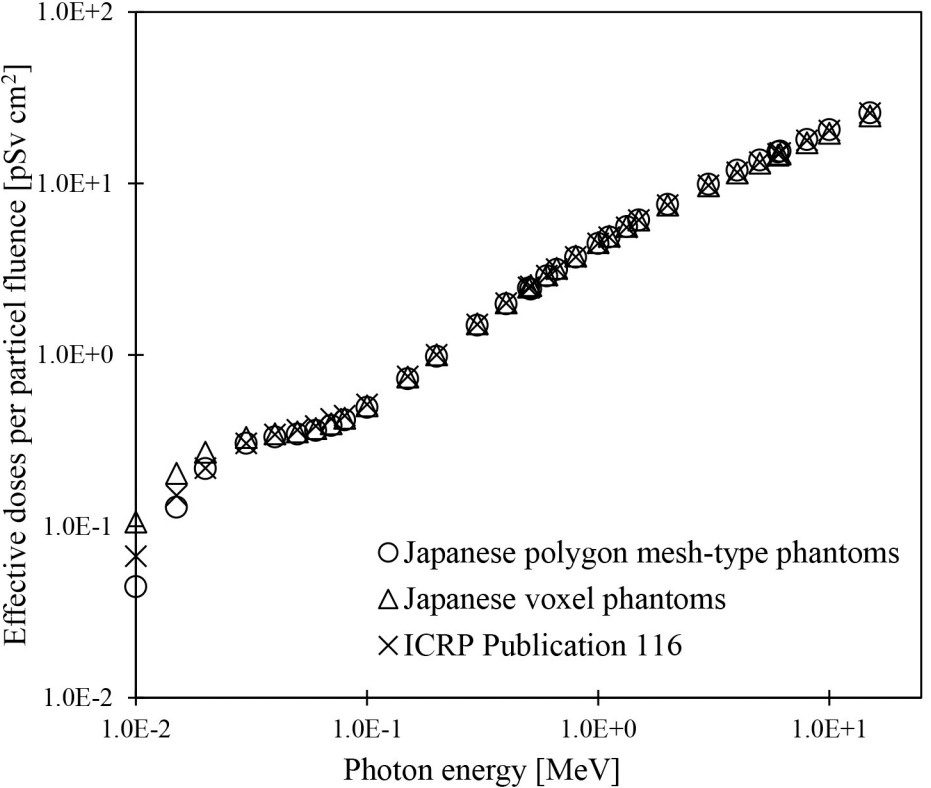

**Fig 4. Effective doses evaluated using Japanese polygon mesh-type phantom (JPM and JPF) and voxel phantoms (JM-103 and JF-103) for photon irradiation in AP geometry and the reference values given in ICRP Publication 116 [4].** The reference effective doses in ICRP Publication 116 were based on the RCP-AM and RCP-AF phantoms.

skin tissue, are in direct contact with the air surrounding the body. Therefore, the breast tissue was directly exposed to the scattered component of primary photon incident on the skin.

Consequently, the effective doses of the voxel phantom were elevated by the increasing equivalent doses of breast tissue with a high tissue weighting factor (S2 Data). In the polygon mesh-type phantoms, the membranous organs have no holes and completely enclose their contents. Therefore, the contribution of the scattered radiation to the equivalent dose of breast tissue is small because of the shielding effect of the skin. A similar trend in effective doses was observed in comparisons with the Japanese polygon mesh-type phantoms and the reference values in ICRP Publication 116 [4]. The ICRP reference values were evaluated using the reference voxel phantoms RCP-AM and RCP-AF [2]. Similar to the comparison between the Japanese polygon mesh-type phantoms and Japanese voxel phantoms, the differences in phantom description formats might cause differences in effective doses between the Japanese polygon mesh-type phantoms and the reference values of the ICRP. These results confirmed that JPM and JPF can accurately evaluate the absorbed doses to membranous organs and their surrounding organs with complicated structures and can be applied to dose assessments against adult Japanese subjects without problems.

## 3.2 Absorbed doses to skin tissues

The differences in skin doses between polygon mesh-type and voxel phantoms were investigated by calculating the absorbed doses to the all-skin regions in JPM and JPF and comparing them with those of JM-103 and JF-103 when these phantoms were irradiated with photons with energy ranges from 0.01 to 20 MeV in AP geometry. As shown in Fig 5 (S2 Table), the doses absorbed in the all-skin regions of the male model, JPM, agreed with those in the JM-103 within 5% over all the calculated energy ranges. Similar results in the female model JPF were compared with JF-103 (S8 Table). These results can be attributed to the following reasons. The macroscopic energy deposition distribution due to photon incidents on the body primarily affects the absorbed doses to the all-skin regions. The microscopic shapes, such as bumps, holes, and discontinuities, found on the skin surface of voxel phantoms are negligible.

Fig 6 compares the doses absorbed in all-skin regions and those in the sensitive regions at a depth of 50–100 μm from the skin surface in the JPM phantom when the irradiation of the photons at energy ranges from 0.01 to 20 MeV was performed in AP geometry (S3 Table). The absorbed doses to the sensitive regions were higher than those in the all-skin regions below 0.015 MeV by 19%–56%. As discussed in ICRP Publication 145 [19], this result was caused because photons with energies of 0.01 and 0.015 MeV have a short mean free path, and their energy deposition significantly increases in the 50–100 μm layer from the skin surface. At energies between 0.02 and 0.3 MeV, the differences in absorbed doses between sensitive regions and all-skin regions were within 10%. At energies above 0.4 MeV, the absorbed doses to the sensitive regions were lower than those to the all-skin regions (2%–36%). Similar results were also obtained from the dose analysis using JPF (S9 Table). The trend observed in this study was attributed to the increased energy deposition resulting from the build-up effects at deeper positions than the sensitive region. The variation characteristics of absorbed doses to the sensitive and all-skin regions in JPM and JPF are similar to those previously reported by ICRP [19].

## 3.3 Absorbed doses to eye tissues

Fig 7 compares the absorbed doses to the entire lens (hereafter referred to as all-lens regions) between JPF and JF-103 (S4 Table). JPF and JF-103 were irradiated with photons with energy ranging from 0.01 to 20 MeV in AP geometry. The maximum difference between JPF and JF-

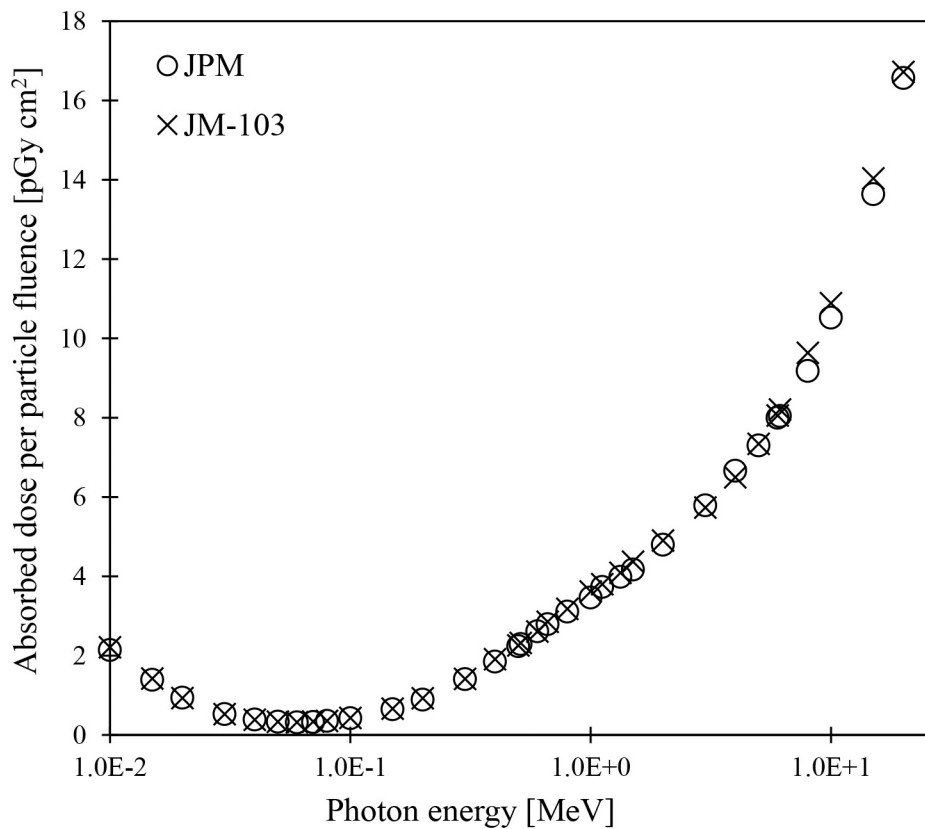

**Fig 5. Absorbed doses to all-skin regions in the JPM and JM-103 phantoms for photon irradiation in AP geometry.**

103 at energy ranges from 0.015 to 20 MeV is approximately 7%, which is in good agreement. Similarly, the absorbed doses to the all-lens regions in JPM were equivalent to those in JM-103 within 7% over the evaluated energy ranges (S10 Table). Regardless of gender, no significant differences were found when comparing the polygon mesh-type phantoms with the voxel phantoms. Although the all-lens region is a very small tissue, the disk-like flat shape and position near the anterior surface of the eye tissue can be easily modeled using voxels of approximately 1 mm³. Thus, the shapes of the all-lens regions in JM-103 and JF-103 do not significantly differ from those of JPM and JPF. Nguyen et al. [37] reported that the absorbed doses to the all-lens regions evaluated using the polygon mesh-type phantoms MRCP-AM and MRCP-AF are generally consistent with those of the voxel phantoms RCP-AM and RCP-AF. The results of this study correlate with those reported by Nguyen et al. [37]. These facts indicate no practical problem with applying JPM and JPF to assess absorbed doses to all-lens regions of adult Japanese.

As described in Subsection 2.2.3, the lens of the detailed eye tissue model [36], which was incorporated into the JPM and JPF phantoms, comprises sensitive and insensitive regions and has a complicated structure. The sensitive and insensitive regions are on the anterior and posterior sides of the lenses, respectively. The absorbed doses to the sensitive and insensitive regions of the lenses in JPM and JPF were calculated, and their dosimetric characteristics were discussed. Fig 8 shows the absorbed doses to the sensitive and insensitive regions of the lens in JPM for irradiating photons with energy ranging from 0.01 to 20 MeV in the AP geometry

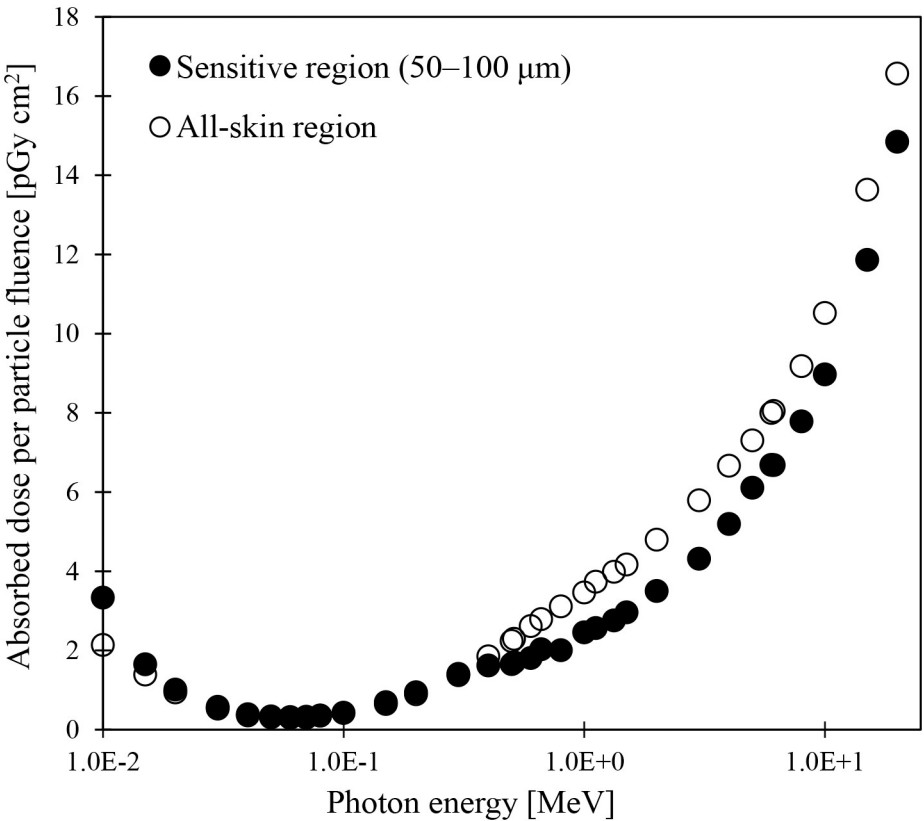

**Fig 6. Absorbed doses to sensitive and all-skin regions in the JPM phantom for photon irradiation in AP geometry.**

(S5 Table). The absorbed doses to the sensitive region were higher than those to the insensitive region in the energy ranging from 0.01 to 0.02 MeV by 8%–70%. At energies between 0.03 and 1.117 MeV, the maximum difference was within 2%. At energies exceeding 1.33 MeV, the absorbed doses to the sensitive region were 2%–27% lower than those to the insensitive region. The differences in variations of absorbed doses between the sensitive and insensitive regions strongly depend on the location of each region in the lens. For AP irradiation conditions, even low-energy photons with short mean free paths can reach and impart energy to the sensitive region, which is a short distance from the eye tissue surface. The reason why the absorbed doses to the insensitive region were higher than those to the sensitive region at energies above 1 MeV is thought to be as follows: At this energy, the build-up peak is behind the eyeball. Therefore, the scattered electrons from the interactions between photons and peripheral tissues behind the eyeball provide a higher absorbed dose in the insensitive region. The above trends in absorbed doses were also observed in JPF with smaller eyeballs than in JPM (S11 Table). These results confirmed the variations in the absorbed doses to eyeballs reported in ICRP Publication 116 [4] and Nguyen et al. [37].

Fig 9 shows the energy dependence of the absorbed doses by the lens and eyeball in the JPF when electrons with energies between 0.03 and 20 MeV were irradiated in the AP geometry (S6 Table). Peaks and slopes of absorbed doses to the sensitive and insensitive regions in the lens were observed at 0.6–1.5 MeV and 0.662–2 MeV, respectively. Although the peak and slope of the absorbed doses to the insensitive region shifted slightly toward high energy

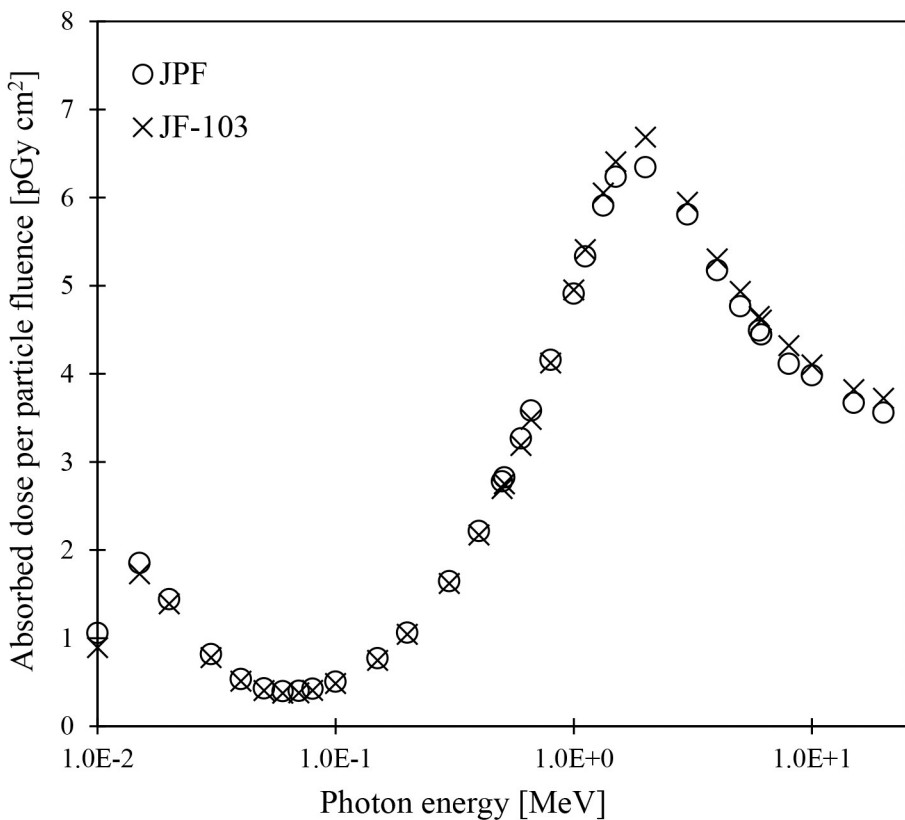

**Fig 7. Absorbed doses to the all-lens regions in the JPF and JF-103 phantoms for photon irradiation in AP geometry.**

compared with those of the sensitive region, no difference occurred in the variation trends in absorbed doses between the sensitive and insensitive regions. Under the irradiation conditions of the AP geometry, these results were attributed to the arrangement of sensitive and insensitive regions in the lens. At energies above 3 MeV, the differences in absorbed doses between the sensitive and insensitive regions were within 2% because the electrons homogeneously reached the entire lens. The absorbed doses to eyeballs were higher than those to the sensitive and insensitive regions in the lens at energies of 0.8 MeV or less. The eyeballs are in contact with the surrounding air. Therefore, direct irradiation of electrons induced an increase in absorbed doses. At energy ranges between 1 and 3 MeV, the doses absorbed by eyeballs were lower than those absorbed by the sensitive and insensitive regions in the lens. As described above, the lens has a disk-like flat shape and is near the anterior surface of the eyeball. The electron energy required to irradiate the entire lens region is lower than the eyeball's. Hence, the doses absorbed by the lens were higher than those of the eyeball. The electrons with energies above 4 MeV have ranges to go through the eyeballs, so the absorbed doses to the eyeballs were similar to those to the lens. The similar energy dependence of the absorbed dose by the lens and eye was also seen in the JPM (S12 Table). These results indicated that distances from the eye tissue surface affect the dose absorbed in the lens's sensitive and insensitive regions and the eyeballs. These results were consistent with those of a report [51] by Furuta et al. At the same energy, the changes in absorbed doses to each lens region due to electrons with short arrival distances from the incidence point are larger than those due to photons (Figs 8 and 9). These results are similar to those reported by Nguyen et al. [37].

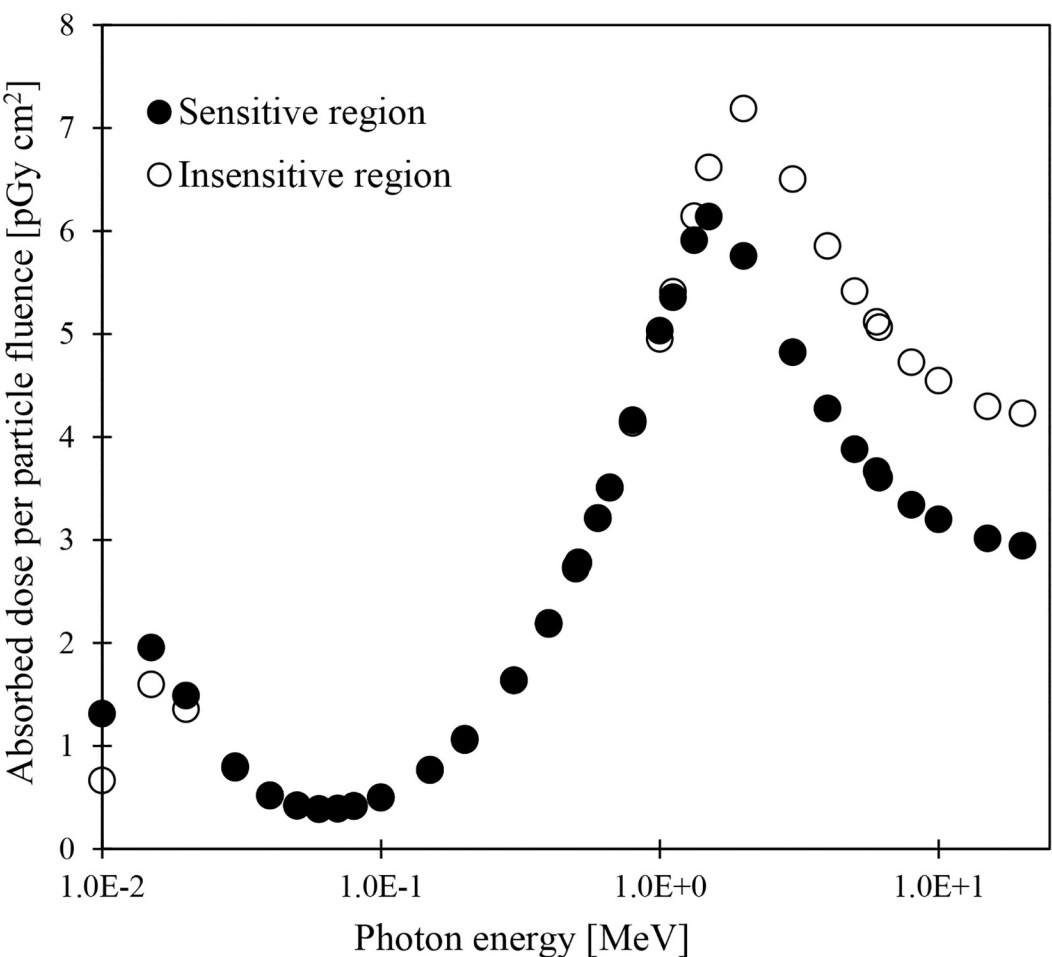

**Fig 8. Absorbed doses to the lens (sensitive and insensitive regions) in the JPM phantom for photon irradiation in AP geometry.**

Fig 10 compares the JPM and JPF phantoms for the absorbed doses to the sensitive region of the lenses and eyeballs for irradiating electrons at 0.03–20 MeV in AP geometry (S7 Table). The slope of the absorbed dose to the sensitive region in the lens for JPM and JPF was seen at 0.6–1.5 MeV. The slope of the absorbed doses in JPM was slightly toward high energy compared with those in JPF. The distances from the eyeball surface to the sensitive region of the lens in JPF are 2.0–3.2 mm, shorter than those (2.1–3.4 mm) in JPM. The differences in the distances between JPM and JPF were attributed to the mass characteristics (Tables 2 and 3) of the lenses and eyeballs. Based on the above, it was considered that the energy of electrons required to reach the sensitive region of the lens was lower in JPF than in JPM, and its absorbed doses to the lens elevated rapidly. Additionally, the eye balls in JPM are more deeply distributed in the head than those in JPF. The lengths in the AP direction of the eyeballs in JPM and JPF are 25.3 and 23.6 mm, respectively. Therefore, for the same energy, more electrons can reach the entire region of the eyeballs in JPF, and larger energies are deposited than in JPM. These results indicated that it was necessary to reproduce the anatomical characteristics of the subjects in detail for accurate evaluations of absorbed doses to the lenses and eyeballs due to external electron exposures. Furthermore, it was also shown that polygon surface

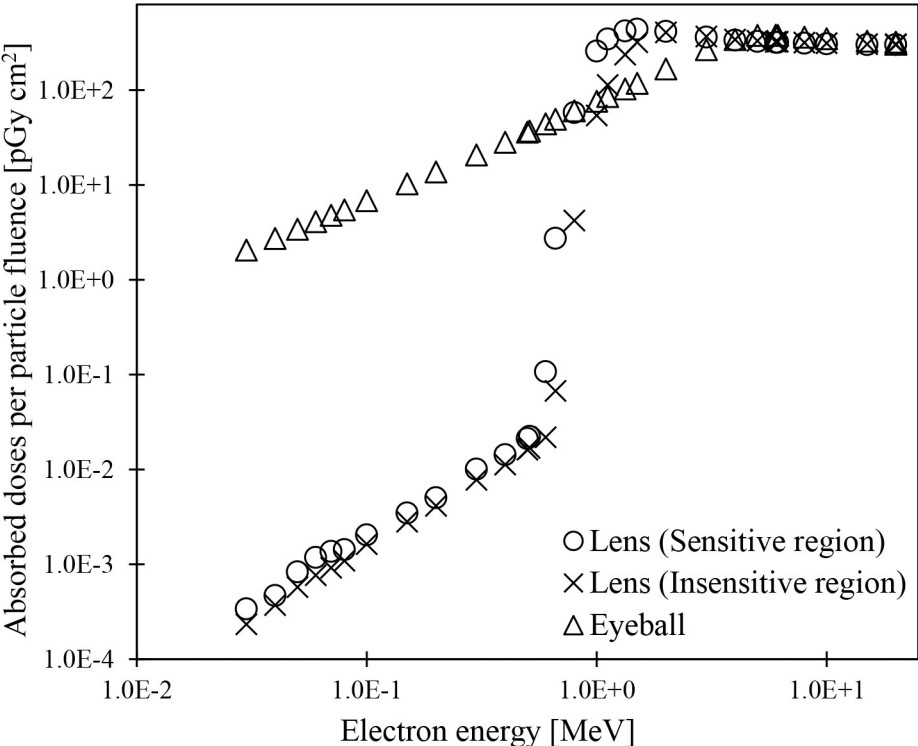

**Fig 9. Absorbed doses to the lens (sensitive and insensitive regions) and eyeball of the JPF phantom for electron irradiation in AP geometry.**

modeling techniques help construct eyeballs and lenses of different sizes depending on the subject.

## 4 Conclusions

We constructed adult JPM and JPF phantoms by applying the polygon surface modeling technique to the Japanese adult voxel phantoms JM-103 (male) and JF-103 (female). Therefore, the masses of organs, tissues, and organ contents in JPM and JPF closely matched the averages of adult Japanese with better precision than JM-103 and JF-103. Thus, the JPM and JPF will be valuable for dose assessments of representative adult Japanese subjects. The phantom data (JPM and JPF) and the input files for implementing these phantoms in PHITS are freely available from the GitHub repository (https://github.com/JapanesePolygonPhantom).

It was confirmed that the JPM and JPF tetrahedral mesh-type data are ready for dose calculations using PHITS. The effective doses of the JPM and JPF phantoms were calculated in the AP geometry for external photon irradiation and compared with those of JM-103 and JF-103 or the reference values [4]. The effective doses by JPM and JPF correlated well with those by JM-103 and JF-103 over most energy ranges. It can be concluded that there are no problems in applying JPM and JPF to dose assessments in adult Japanese subjects. Furthermore, the JPM and JPF phantoms can realistically reproduce the shape of membranous tissue, such as skin, which was challenging with the JM-103 and JF-103 phantoms because of the limitations of resolution and geometry in a voxel format.

The absorbed doses to the entire skin and lens tissue of JPM and JPF correlated well with those of JM-103 and JF-103, respectively. In dose analysis using the JPM and JPF phantoms,

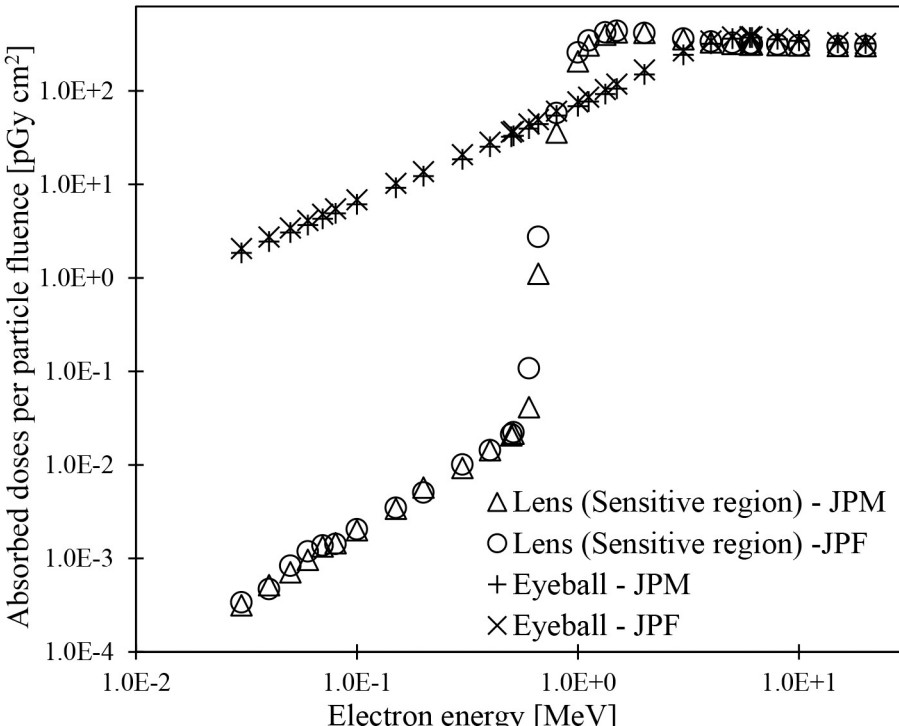

**Fig 10. Comparison of the absorbed doses to lens (sensitive region) and eyeball in JPM with those in JPF for electron irradiation in AP geometry.**

the absorbed doses to sensitive regions of the skin and lens differ from those to the entire region. These results were caused by differences in the geometries between the entire region and sensitive regions in the skin and lens tissue. Thus, these results indicated that JPM and JPF are valuable for dose assessments of sensitive regions with thin, small, and complicated shapes, such as the skin and lens. Therefore, it was shown that the JPM and JPF can provide more accurate exposure dose calculation than the JM-103 and JF-103 phantoms.

We are currently developing postural deformation techniques for JPM and JPF. Together with posture deformation techniques, JPM and JPF will become powerful evaluation tools for exposure doses, considering each exposed individual's posture and body size in dose assessments for medical treatments and radiation accidents.

## Supporting information

**S1 Table. The numerical values plotted in Fig 4.**
(XLSX)

**S2 Table. The numerical values plotted in Fig 5.**
(XLSX)

**S3 Table. The numerical values plotted in Fig 6.**
(XLSX)

**S4 Table. The numerical values plotted in Fig 7.**
(XLSX)

**S5 Table. The numerical values plotted in Fig 8.**
(XLSX)

**S6 Table. The numerical values plotted in Fig 9.**
(XLSX)

**S7 Table. The numerical values plotted in Fig 10.**
(XLSX)

**S8 Table. Absorbed doses to all-skin regions in the JPF and JF-103 phantoms for photon irradiation in AP geometry.**
(XLSX)

**S9 Table. Absorbed doses to sensitive and all-skin regions in the JPF phantom for photon irradiation in AP geometry.**
(XLSX)

**S10 Table. Absorbed doses to the all-lens regions in the JPM and JM-103 phantoms for photon irradiation in AP geometry.**
(XLSX)

**S11 Table. Absorbed doses to the lens (sensitive and insensitive regions) in the JPF phantom for photon irradiation in AP geometry.**
(XLSX)

**S12 Table. Absorbed doses to the lens (sensitive and insensitive regions) and eyeball of the JPM phantom for electron irradiation in AP geometry.**
(XLSX)

**S1 Data. The absorbed doses to active marrow and endosteum calculated by using Japanese polygon mesh-type (male: JPM, female: JPF) and voxel-type (male: JM-103, female: JF-103) phantoms for irradiation of photon with energies from 0.01 to 20 MeV in AP geometry, along with the reference values (ICRP Publication 116) based on RCP-AM and RCP-AF.**
(XLSX)

**S2 Data. The absorbed doses, equivalent doses calculated by using JPM, JPF, JM-103 and JF-103, along with contributions of equivalent doses to effective doses for 0.01, 0.015, 0.02 and 0.03MeV photon irradiation in AP geometry.**
(XLSX)

## Acknowledgments

The authors would like to express our gratitude to Dr F Takahashi of Nuclear Emergency Assistance and Training Center, Japan Atomic Energy Agency for his helpful and valuable advice on the present work.

## Author Contributions

**Conceptualization:** Kaoru Sato, Takuya Furuta, Daiki Satoh, Shuichi Tsuda.

**Investigation:** Kaoru Sato, Takuya Furuta.

**Methodology:** Kaoru Sato, Takuya Furuta, Daiki Satoh, Shuichi Tsuda.

**Validation:** Kaoru Sato.

**Writing – original draft:** Kaoru Sato.

**Writing – review & editing:** Kaoru Sato, Takuya Furuta, Daiki Satoh, Shuichi Tsuda.

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
