## [Decision Letter · Decision Letter 0]

25 Jun 2024

PONE-D-24-12142Construction of new polygon mesh-type phantoms based on adult Japanese voxel phantomsPLOS ONE

Dear Dr. Sato,

Thank you for submitting your manuscript to PLOS ONE. After careful consideration, we feel that it has merit but does not fully meet PLOS ONE’s publication criteria as it currently stands. Therefore, we invite you to submit a revised version of the manuscript that addresses the points raised during the review process.

We look forward to receiving your revised manuscript.

Kind regards,

Sakae Kinase, Ph.D.

Academic Editor

PLOS ONE

Additional Editor Comments:

This paper has been carefully considered by two referees. One referee has a positive opinion of the paper, while the other has commented on some important issues-treatments of blood and skeleton in phantom developments. The comments indicate that some fundamental revisions are necessary before the paper can again be considered for publication in PLOS ONE. Please revise your manuscript in accordance with reviewers' comments.

Reviewers' comments:

Reviewer's Responses to Questions

**Comments to the Author**

1. Is the manuscript technically sound, and do the data support the conclusions?

Reviewer #1: Yes

Reviewer #2: Yes

2. Has the statistical analysis been performed appropriately and rigorously? 

Reviewer #1: Yes

Reviewer #2: Yes

3. Have the authors made all data underlying the findings in their manuscript fully available?

Reviewer #1: Yes

Reviewer #2: No

4. Is the manuscript presented in an intelligible fashion and written in standard English?

Reviewer #1: Yes

Reviewer #2: Yes

5. Review Comments to the Author

Reviewer #1: PONE-D-24-12142

Construction of new polygon mesh-type phantoms based on adult Japanese voxel phantoms

Sato et al.

General Comments

This study sought to perform a voxel-to-mesh conversion of the existing voxel-type Japanese reference phantoms to a mesh-type format. The process is well described with a few exceptions – (1) issue of blood content in the organs, and (2) issue of the physical separation of cortical bone and spongiosa within the skeletal models. The utility of the new Japanese mesh-type phantoms is highlighted in a series of comparisons of organ and effective dose coefficients for idealized irradiation geometries as described in ICRP Publication 116. This is a tremendous effort, and the paper is very well written and presented. The effort effectively mirrors that of ICRP Task Group 103 in the similar conversion of phantom formats in the ICRP adult male and adult female reference phantoms. Important issues requiring further clarification are listed below.

Specific Comments

Page 2, Line 56

Change “modeled adult Japanese polygon mesh-type male” to “created adult Japanese polygon mesh-type male”

Change “by modifying the” to “through modification of the”

Page 3, Section 2.1

Please give some background information on the reference or mean values of organ masses and volumes for the current Japanese pollution – as used as target volumes in both the previous voxel-type adult male and adult female phantoms, and the newly created mesh-type phantoms. As the authors are well aware, the organs within the ICRP Publication 110 voxel-type reference phantoms are undersized as the reference masses in ICRP Publication 89 were presumed to be inclusive of organ blood volume/mass while in fact they were exclusive of blood volume/mass. As such, a major effort in ICRP Publication 145 was the increase in organ masses from the ICRP Publication 110 phantoms in the voxel-to-mesh conversion project.

Does this same issue reside in the Japanese reference phantoms – either or both the voxel-type or mesh-type versions? There are no data tables of organ masses and so it is unclear whether or not reference values exist for the Japanese reference phantoms that distinguish organ tissue mass from organ blood mass. Please elaborate in this section or elsewhere as readers of your paper will be well aware of this issue from ICRP Publications 110 and 145.

Page 5, Section 2.2.6

From this section of the study methods, it appears that there was no attempt at differentiating within the skeleton of the Japanese reference phantoms – cortical bone and trabecular spongiosa. Are the skeleton models in both the voxel-type and mesh-type homogenous with respect to cortical bone and spongiosa? If this is true, this is a significant limitation of the phantoms as external and internal radiation doses to both active marrow and bone endosteum – which only reside in the spongiosa regions of the skeleton – will be overestimated as they do not account the particle shielding effect of cortical bone. Please explain if this is the case, and why an attempt to differentiate cortical bone in the mesh-type phantoms was not attempted.

Page 8, Table 4 Caption

Please move this to the top of page 9

Page 10, Section 2.4

If the skeleton of the mesh-type phantoms is homogeneous and does not include a separate tetrahedral region of cortical bone, then this of course impacts the dose equations on page 10. It would be helpful to take the ICRP Publication 145 phantoms and homogenize that skeleton to match the homogeneous skeleton of the Japanese mesh-type phantom and then perform a sensitivity study of the importance of cortical bone shielding – especially for low-energy photons incident upon the skeletal regions.

General Comment

This manuscript should additionally include 3D images of both the Japanese voxel-type and Japanese mesh-type phantoms. Please see Figures 2.1 and Figures 6.1/6.2 in ICRP Publication 145. A similar set of images would greatly enhance this paper.

Reviewer #2: Excellent work, thanks for sharing this interesting and valuable research!

Just two (minor) comments:

Lines 507-508: the Table S12 does not show differences in male/female response, as the sentence would suggest

Lines 124 and 125 (and also 547 and 548): no "phantom data (JPM and JPF) and the input files for implementing phantoms in PHITS" are loaded in the GitHub repository yet.

I also read, in page 5, that the plan is to publish them after acceptance of the article.

6. PLOS authors have the option to publish the peer review history of their article (what does this mean?). If published, this will include your full peer review and any attached files.

Reviewer #1: No

Reviewer #2: **Yes: **Daniele Giuffrida

---

## [Author Response · Author response to Decision Letter 0]

6 Aug 2024

PLOS ONE

Academic Editor

Sakae Kinase, Ph.D.

Dear Dr. Sakae Kinase,

Additional Editor Comments:

This paper has been carefully considered by two referees. One referee has a positive opinion of the paper, while the other has commented on some important issues-treatments of blood and skeleton in phantom developments. The comments indicate that some fundamental revisions are necessary before the paper can again be considered for publication in PLOS ONE. Please revise your manuscript in accordance with reviewers' comments. 

Reply of author to Additional Editor Comments:

 Thank you very much for your e-mail of June 25, 2024, with regard to our manuscript entitled “Construction of new polygon mesh-type phantoms based on adult Japanese voxel phantoms” (manuscript ID: PONE-D-24-12142) together with the comments from the reviewers. We tried to revise the manuscript as much as possible in line with the suggestions made by you and the reviewers. In the revised manuscript, the characteristics and issues of bone tissue structure, and the relationship between organ mass and blood content in organ of the newly developed Japanese polygon mesh-type phantom have been reinforced and improved. The authors submitted a rebuttal letter that responds to you and reviewers ('Response to Reviewers-PONE-D-24-12142''), the marked-up manuscript that highlights changes made to the original submitted version ('Revised Manuscript with Track Changes-PONE-D-24-12142') and the revised paper without tracked changes ('Manuscript-PONE-D-24-12142'). We appreciate the helpful suggestions offered by the you and reviewers as these comments were very useful for revising this manuscript. We believe the manuscript has been improved satisfactorily and hope it will be accepted for publication in PLOS ONE. 

Sincerely yours, 

Kaoru Sato

Nuclear Science and Engineering Center, 

Japan Atomic Energy Agency

Reviewer #1: PONE-D-24-12142

Construction of new polygon mesh-type phantoms based on adult Japanese voxel phantoms

Sato et al.

General Comments

This study sought to perform a voxel-to-mesh conversion of the existing voxel-type Japanese reference phantoms to a mesh-type format. The process is well described with a few exceptions – (1) issue of blood content in the organs, and (2) issue of the physical separation of cortical bone and spongiosa within the skeletal models. The utility of the new Japanese mesh-type phantoms is highlighted in a series of comparisons of organ and effective dose coefficients for idealized irradiation geometries as described in ICRP Publication 116. This is a tremendous effort, and the paper is very well written and presented. The effort effectively mirrors that of ICRP Task Group 103 in the similar conversion of phantom formats in the ICRP adult male and adult female reference phantoms. Important issues requiring further clarification are listed below. 

Reply of author to comment:

 Thank you very much for your very evaluate comments to our manuscript. In particular, as you pointed out, (1) issue of blood content in the organs, and (2) issue of the physical separation of cortical bone and spongiosa within the skeletal models were not clear in submitted our paper. Therefore, we tried to revise the manuscript as much as possible in line with the suggestions made by you. The responses to the reviewer's specific comments are as follows:

Specific Comments

Page 2, Line 56

Change “modeled adult Japanese polygon mesh-type male” to “created adult Japanese polygon mesh-type male” 

Reply of author to reviewer comment: 

 “modeled adult Japanese polygon mesh-type male” has been substituted with “created adult Japanese polygon mesh-type male” as suggested (Page 2, Line 56).

Change “by modifying the” to “through modification of the”

Reply of author to reviewer comment:

 “by modifying the” has been changed to “through modification of the” as suggested (Page 2, Lines 56-57).

Page 3, Section 2.1

Please give some background information on the reference or mean values of organ masses and volumes for the current Japanese pollution – as used as target volumes in both the previous voxel-type adult male and adult female phantoms, and the newly created mesh-type phantoms. As the authors are well aware, the organs within the ICRP Publication 110 voxel-type reference phantoms are undersized as the reference masses in ICRP Publication 89 were presumed to be inclusive of organ blood volume/mass while in fact they were exclusive of blood volume/mass. As such, a major effort in ICRP Publication 145 was the increase in organ masses from the ICRP Publication 110 phantoms in the voxel-to-mesh conversion project. 

Reply of author to reviewer comment:

 Thank you for your valuable comments. The Japanese averages of organ masses and body sizes adopted for the mesh-type phantoms (JPM and JPF) and voxel phantoms (JM-103 and JF-103) are based on Japanese anatomical data reported by Tanaka and Kawamura (1996). The reported values by Tanaka and Kawamura (1996) were derived from data on healthy individuals whose cause of death was determined to be sudden death by autopsies performed at the Tokyo Metropolitan Medical Examiner's Office. In the autopsies described above, the masse of organs and blood in organs were measured separately; the average organ masses for Japanese reported by Tanaka and Kawamura (1996) are the total mass of organ and blood in organs. In other words, the Japanese average of organ masses adopted by this study includes blood. The above point was not clear in the submitted paper, and have been added in Section 2.1 (Page 3, Lines 129-136).

Does this same issue reside in the Japanese reference phantoms – either or both the voxel-type or mesh-type versions? There are no data tables of organ masses and so it is unclear whether or not reference values exist for the Japanese reference phantoms that distinguish organ tissue mass from organ blood mass. Please elaborate in this section or elsewhere as readers of your paper will be well aware of this issue from ICRP Publications 110 and 145. 

Reply of author to reviewer comment:

 The organ masses by Tanaka and Kawamura (1996), which adopted as the averages for Japanese people, include the mass of blood in the organs. The organ masses of the mesh-type phantoms (JPM and JPF) and voxel phantoms (JM-103 and JF-103) were adjusted to the Japanese average (Tanaka and Kawamura, 1996) including the mass of blood in organs. Therefore, the problems about organ masses and blood in organs of the reference voxel phantoms of ICRP Publication 110 and the reference polygon mesh-type phantoms of ICRP Publication 145 does not exist for the Japanese phantoms (JPM, JPF, JM-103, and JF-103). The above point was not clear in the submitted paper, so I have been added in section 2.3 (Page 7, Lines 297-301). In addition, it has been added to footnote of Figs 2 and 3 that the organ and tissue masses of Japanese polygon meshy-type and voxel-type phantoms, and the Japanese averages include the blood in the organs and tissues (Pages 7-9).

Page 5, Section 2.2.6

From this section of the study methods, it appears that there was no attempt at differentiating within the skeleton of the Japanese reference phantoms – cortical bone and trabecular spongiosa. Are the skeleton models in both the voxel-type and mesh-type homogenous with respect to cortical bone and spongiosa? If this is true, this is a significant limitation of the phantoms as external and internal radiation doses to both active marrow and bone endosteum – which only reside in the spongiosa regions of the skeleton – will be overestimated as they do not account the particle shielding effect of cortical bone. Please explain if this is the case, and why an attempt to differentiate cortical bone in the mesh-type phantoms was not attempted. 

Reply of author to reviewer comment:

 Thank you for your important comment. To reproduce the non-uniform distributions and structures of hard bone and bone marrow within bone tissue, 20 anatomical bone regions (e.g. lumbar vertebrae, ribs, etc.) of the voxel-type phantoms (JM-103 and JF-103) were segmented into seven material regions with different bone marrow contents, densities, and elemental compositions. In other words, the bone tissues of JM-103 and JF-103 were segmented into 140 regions (7 materials × 20 anatomical bone regions) over the whole body. Each anatomical bone region of JM-103 and JF-103 has a non-uniform density distribution (Sato et al., JAEA-Data/Code 2011–013). The bone density distribution of each anatomical bone region was evaluated by using the CT image data (resolution: 0.98 x 0.98 x 1mm3) of actual persons employed as volunteers for phantom construction. The complex internal structure of bone tissue including cortical bone, spongiosa and marrow cavity are approximately represented as the density distribution in bone region by using the segmentation technique described above. 

 The anatomical bone regions of the Japanese polygon mesh-type phantoms (JPM and JPF) are not segmented into the cortical bone, spongiosa and marrow cavity. The reason is as follows. We are currently developing a function for changing posture and body size of JPM and JPF, which is important for evaluating individual radiation doses. The posture and body size changing function under development have the goal of simultaneously deforming the shape of the anatomical bone region and internal structures (cortical bone, trabecular bone, medullary cavity). Since the basic bone internal structure of cortical bone, spongiosa and marrow cavity that can accommodate changes in posture and body size of JPM and JPF has not been determined, the errors which are obstacles to tetrahedral element generations and radiation transport calculations, occur. The basic bone internal structures in each anatomical bone regions are under construction. We will incorporate the basic bone internal structures into to the anatomical bone regions of JPM and JPF. As you pointed out, the segmentation of the internal structure (cortical bone, spongiosa and marrow cavity) of bone tissue is important for accurately assessing the exposure doses to active marrow and endosteum. We think so, too. In the submitted paper, it was not clear that the internal structures of bone tissues are not segmented in JPM and JPF, and the reason for the lack of segmentation of the internal structure was also not explained. Therefore, the limitations of using JPM and JPF to evaluate the exposure doses to active marrow and endosteum were also not clear. I have been added the explanations to Subsection 2.2.6 (Pages 5-6, Lines 250-272) and Section 2.4 (Page 9, Lines 321-337) of the submitted paper to avoid misunderstandings among readers.

Page 8, Table 4 Caption

Please move this to the top of page 9

Reply of author to reviewer comment:

 I have been moved the caption for Table 4 to the top of page 10, paying attention to its position on the page (Page 10, Top).

Page 10, Section 2.4

If the skeleton of the mesh-type phantoms is homogeneous and does not include a separate tetrahedral region of cortical bone, then this of course impacts the dose equations on page 10. It would be helpful to take the ICRP Publication 145 phantoms and homogenize that skeleton to match the homogeneous skeleton of the Japanese mesh-type phantom and then perform a sensitivity study of the importance of cortical bone shielding – especially for low-energy photons incident upon the skeletal regions. 

Reply of author to reviewer comment:

 Thank you for the most useful advice. In each anatomical region of JM-103 and JF-103, the complex internal structures including cortical bone, spongiosa and marrow cavity are approximately represented as the density distribution in bone region. On the other hand, in each anatomical bone region of JPM and JPF, the complex internal structures are not segmented and have a uniform density distribution. In addition, the elemental composition and density of entire each anatomical bone region (e.g., entire cranium) of JPM are adjusted to be equal to those of JM-103 (JPF is also adjusted to be equal to JF-103). Therefore, in this revised submission paper, a sensitivity analysis of the impact of the complex internal structures in bone regions on absorbed dose to active marrow or endosteum have been added (Pages 11-12, Lines 368-420). In sensitivity analysis, the absorbed doses to the active marrow or endosteum of JPM and JPF with uniform density distribution in each anatomical bone region were compared with those of JM-103, JF-103, RCP-AM and RCP-AF which have non-uniform density distribution in each anatomical bone region, when photons with energy ranges from 0.01 to 20 MeV in AP geometry were irradiated. The calculation result data on which these discussions are based, is presented in S1 Data (Page 11, Line 386; Page 12, Lines 398-399; Page 12, Line 417). 

By adding S1 data, a caption for the S1 data has been added (Page 20, Lines 852-855). 

By adding S1 Data, the previous S1 Data has been changed to S2 Data (Page 13, Line 496; Page 20, Line 857). 

Revisions and additions were made to the descriptions as suggested. As a result, the numbering of references [47] and [48] was reversed. That is, the previous reference [47] has been renumbered [48], and the previous reference [48] has been renumbered [47] (Page 19, Lines 806-810).

General Comment

This manuscript should additionally include 3D images of both the Japanese voxel-type and Japanese mesh-type phantoms. Please see Figures 2.1 and Figures 6.1/6.2 in ICRP Publication 145. A similar set of images would greatly enhance this paper.

Reply of author to reviewer comment:

 Thank you for your valuable comments. As you suggested, I have prepared figures for the Japanese phantoms of voxel type (JM-103 and JF-103) and mesh-type (JPM and JPF). The figures for JM-103 and JF-103 as Fig 1 has been added (Page 3, Lines 139-140, Lines 152-154). In addition, the figures for JPM and JPF as Figs 2 and 3 have been added, respectively (Page 5, Lines 221, 224 and 227). Therefore, the numbers in the following figures have been changed as follows:

By adding Figs 1, 2, and 3, the previous Fig 1 has been changed to Fig 4 (Page 13, Lines 473 and 488; Page 20, Line 823).

By adding Figs 1, 2, and 3, the previous Fig 2 has been changed to Fig 5. (Page 14, Lines 513 and 522; Page 20, Line 825).

By adding Figs 1, 2, and 3, the previous Fig 3 has been changed to Fig 6 (Page 14, Lines 526 and 541; Page 20, Line 827).

By adding Figs 1, 2, and 3, the previous Fig 4 has been changed to Fig 7 (Page 14, Line 546; Page 15, Line 563: Page 20, Line 829).

By adding Figs 1, 2, and 3, the previous Fig 5 has been changed to Fig 8 (Page 15, Lines 570 and 588; Page 20, Line 831).

By adding Figs 1, 2, and 3, the previous Fig 6 has been changed to Fig 9 (Page 15, Line 592; Page 16, Line 618; Page 20, Line 833). 

By adding Figs 1, 2, and 3, the previous Figs 5 and 6 has been changed to Figs 8 and 9 (Page 16, Line 615).

By adding Figs 1, 2, and 3, the previous Fig 7 has been changed to Fig 10 (Page 16, Lines 622 and 640; Page 20, Line 835).

Reviewer #2: Excellent work, thanks for sharing this interesting and valuable research!

Just two (minor) comments:

Lines 507-508: the Table S12 does not show differences in male/female response, as the sentence would suggest 

Reply of author to reviewer comment:

 Thank you for accurate comment. The original sentence in submitted was inappropriate. Therefore, the sentence "No gender differences were found in the above results." has been changed to "The similar energy dependence of the absorbed dose by the lens and eye was also seen in the JPM." (Page 16, Lines 609-610).

Lines 124 and 125 (and also 547 and 548): no "phantom data (JPM and JPF) and the input files for implementing phantoms in PHITS" are loaded in the GitHub repository yet.

I also read, in page 5, that the plan is to publish them after acceptance of the a

---

## [Decision Letter · Decision Letter 1]

19 Aug 2024

Construction of new polygon mesh-type phantoms based on adult Japanese voxel phantoms

PONE-D-24-12142R1

Dear Dr. Sato,

We’re pleased to inform you that your manuscript has been judged scientifically suitable for publication and will be formally accepted for publication once it meets all outstanding technical requirements.

Kind regards,

Sakae Kinase, Ph.D.

Academic Editor

PLOS ONE

Additional Editor Comments (optional):

I have much pleasure in recommending this paper for publication. There are some comments from reviewer 2. Before publication, the authors require appropriate responses to the comments.

Reviewers' comments:

Reviewer's Responses to Questions

**Comments to the Author**

1. If the authors have adequately addressed your comments raised in a previous round of review and you feel that this manuscript is now acceptable for publication, you may indicate that here to bypass the “Comments to the Author” section, enter your conflict of interest statement in the “Confidential to Editor” section, and submit your "Accept" recommendation.

Reviewer #1: All comments have been addressed

Reviewer #2: All comments have been addressed

2. Is the manuscript technically sound, and do the data support the conclusions?

Reviewer #1: Yes

Reviewer #2: Yes

3. Has the statistical analysis been performed appropriately and rigorously? 

Reviewer #1: Yes

Reviewer #2: Yes

4. Have the authors made all data underlying the findings in their manuscript fully available?

Reviewer #1: Yes

Reviewer #2: Yes

5. Is the manuscript presented in an intelligible fashion and written in standard English?

Reviewer #1: Yes

Reviewer #2: Yes

6. Review Comments to the Author

Reviewer #1: The authors were very responsive to the prior review of the manuscript. xxxxxxxxxxxxxxxxxxxxxxxxxxxxxxxxxxxxxxxxxxxxxxxxxxxxxxxxxxxxxxxxxxxxxxxxxxxxxxxxxxxxxxxxxxxxxxxxxxxxxxxxxxxxxxxxxxxxxxxxxxxxxxxxxxxxxxxxxxxxxxxxxx

Reviewer #2: The revised article covers and replies to all the observations that have been raised by the Reviewers in a very satisfactory way.

Modifications included:

1. clarifications on the use of autopsy sudden death cases, which help to understand how widely applicable to the Japanese population the present models are;

2. Figures 1, 2 and 3 provide a visual and very effective representation of the two different models;

3. clarifications (249, 327) on the deformations of the models due to posture adjustments explain why basic bone internal structure of cortical bone, spongiosa and marrow cavity have not been added yet at this stage;

4. clarifications (299, have been added regarding the mass of blood in organs (included);

5. extensive clarifications (371) have been added on the 10% difference in the absorbed doses to active marrow and endosteum between these and the Japanese phantoms, due to different modelling of internal structure of anatomical bone regions, and additional references (818)

With these modifications and explanations, I suggest to accept the paper.

Thank you.

7. PLOS authors have the option to publish the peer review history of their article (what does this mean?). If published, this will include your full peer review and any attached files.

Reviewer #1: **Yes: **Wesley E. Bolch

Reviewer #2: **Yes: **Daniele Giuffrida

---

## [Editor Report · Acceptance letter]

27 Aug 2024

PONE-D-24-12142R1 

PLOS ONE

Dear Dr. Sato, 

I'm pleased to inform you that your manuscript has been deemed suitable for publication in PLOS ONE. Congratulations! Your manuscript is now being handed over to our production team.

Kind regards, 

on behalf of

Professor Sakae Kinase 

Academic Editor

PLOS ONE